# QUANTIFYING VARIANCE IN EVALUATION BENCHMARKS

## ABSTRACT

Evaluation benchmarks are the cornerstone of measuring capabilities of large language models (LLMs), as well as driving progress in said capabilities. Originally designed to make claims about capabilities (or lack thereof) in fully pretrained models, evaluation benchmarks are now also extensively used to decide between various training choices. Despite this widespread usage, we rarely quantify the variance in our evaluation benchmarks, which dictates whether differences in performance are meaningful. Here, we define and measure a range of metrics geared towards measuring variance in evaluation benchmarks, including seed variance across initialisations, and monotonicity during training. By studying a large number of models – both openly available and pretrained from scratch – we provide empirical estimates for a variety of variance metrics, with considerations and recommendations for practitioners. We also evaluate the utility and tradeoffs of continuous versus discrete performance measures and explore options for better understanding and reducing this variance. We find that simple changes, such as framing choice tasks (like MMLU) as completion tasks, can often reduce variance for smaller scale ($\sim$7B) models, while more involved methods inspired from human testing literature (such as item analysis and item response theory) struggle to meaningfully reduce variance. Overall, our work provides insights into variance in evaluation benchmarks, suggests LM-specific techniques to reduce variance, and more generally encourages practitioners to carefully factor in variance when comparing models.

## 1 INTRODUCTION

Evaluation benchmarks are the cornerstone of establishing and defining progress with large language models (LLMs). Virtually any new model release is accompanied by a range of scores on common evaluation benchmarks, illustrating how the model tallies up against previous releases (Mesnard et al., 2024; AI@Meta, 2024; Achiam et al., 2023; Reid et al., 2024). As such, evaluation benchmarks play an important role in claiming progress and the title of state-of-the-art. Consequently, choices in model development are often based on how they impact performance on benchmarks considered important by the field, giving benchmarks a prominent role in model iteration as well. Yet, despite their importance, benchmark scores are often regarded as a point estimate, and it is rare that they are given a more detailed consideration. While it is well known that benchmarks scores can be heavily influenced by the choice of prompt (Sclar et al., 2023), the distributions of labels in the provided few-shots (Weber et al., 2023) or even the symbols that are used for the different options in a multiple choice setup (Zheng et al., 2023; Alzahrani et al., 2024), papers rarely report more than a single number per benchmark, or specifics on how each number was computed. Furthermore, statistical significance values are scarcely reported on major release papers or leaderboards, or even in papers that study how scores vary across various dimensions. These issues muddy the power of evaluation benchmarks, both during development and evaluation: if we cannot 'trust' our evaluation results or do not understand what improvements are statistically significant, we cannot make sound comparisons, thus making it more challenging to reliably use benchmarks during model development.

To address this, we present a deep dive into variance in benchmark scores, at much larger scale than any previous work. Across all our experiments, we consider 13 different popular benchmarks and compute their performance over 280 different models, including fully trained public models as well

as a set of 7B models and their intermediate checkpoints that we trained from scratch, differing only in their initialisation random seed.

With this, our contributions are three-fold:

1. We provide a comprehensive reference guide for what magnitudes of variance are expected for what benchmarks across various circumstances.
2. We make suggestions of how variance can be reduced for smaller scale models on choice tasks of important value (MMLU).
3. We caution against the use of efficient benchmarking methods like item analysis and item response theory as a means of reducing cost when doing pre-training ablations, as the methods often lead to increased variance (and thus less power in comparisons as compared to using the full benchmark).

Our work brings to light the often overlooked problem of variance in evaluation benchmarks, quantifies its effects, and provides a set of positive and negative results on how to mitigate it.

## 2 MODELS AND BENCHMARKS

We run our analysis by comparing benchmark results across a large number of models trained across various setups. In this section, we describe these models and list the benchmarks we investigate.

**Models**    We use over 280 models for our analysis, including intermediate checkpoints. First, we train ten Llama-2-7B-architecture models from scratch on our own pre-training data mixture inspired by Touvron et al. (2023a) (See Appendix A). These 10 runs are identical, except for the model initialisation seed. The model hyper-parameters, the pre-training data mixture, and the data-loading mechanism is consistent across all these ten runs. We train these models for 210 billion tokens and store 21 checkpoints for each model, leaving us with 10 sets of 21 model snapshots. We refer to these 210 checkpoints as the "seed models." In addition, we use 41 intermediate and fully-trained models based on the Llama-1 and Llama-2 architecture pre-trained on the same data mixture used for training the seed models.

Finally, we use 32 publicly available models from Huggingface (Wolf et al., 2020): Meta-Llama-3 {8, 70}B (Dubey et al., 2024), Gemma {2, 7}B (Mesnard et al., 2024), DBRX-Base (Databricks, 2024), Mistral 7B (Jiang et al., 2023), Mixtral 8x{7, 22}B (Jiang et al., 2024), Qwen-1.5 {0.5, 1.8, 4, 7, 14, 32, 72, 110}B (Bai et al., 2023), Pythia {1, 1.4, 2.8, 6.9, 12}B (Biderman et al., 2023), Falcon {7, 40}B (Almazrouei et al., 2023), DeepSeek {7, 67}B (Bi et al., 2024), DeepSeek-MoE 16B (Bi et al., 2024), DeepSeek V2 (DeepSeek-AI, 2024), StableLM {1.6, 3, 7}B (StabilityAI, 2024), and MPT {7, 30}B (MosaicML NLP Team, 2023).

The set of models used for the analysis are diverse across architectures, data mixtures, and sizes ranging from 0.5B to 236B total parameters. Details of all models are presented in Table 6.

**Benchmarks**    We do a comprehensive analysis using 13 large-scale well-established NLP benchmarks: AGIEval (Zhong et al., 2023), AI2 Reasoning Challenge (ARC-C) (Clark et al., 2018), BIG Bench (Hard) (Srivastava et al., 2022; Suzgun et al., 2022), COPA (Roemmele et al., 2011), GSM8k (Cobbe et al., 2021), Hellaswag (Zellers et al., 2019), HumanEval (Chen et al., 2021), MATH (Hendrycks et al., 2021), MMLU (Hendrycks et al., 2020), Natural Questions (Kwiatkowski et al., 2019), PIQA (Bisk et al., 2020), SIQA (Sap et al., 2019), and TriviaQA (Joshi et al., 2017).

These benchmarks are a mix of choice- and generation-based benchmarks, that span various capabilities ranging from general knowledge to coding.

## 3 HOW MUCH VARIANCE DO WE OBSERVE?

We first investigate how much variance there is across different models and datasets. We define a range of metrics for quantifying different kinds of variance.

First, using the 7B models we trained ourselves, we consider variance due to changes in seed, across otherwise identical setups. This *seed variance* gives us a metric useful for performing data ablations

– to conclude that pretraining dataset or hyperparameter set B is better than pretraining dataset or hyperparameter set A, we would want the benchmark performance increase to be larger than that due to random seed variance across different models trained in setup A. To this end, we also compute a benchmark's *monotonicity*, quantifying how stably performance on it develops during training.

To ground the seed variance numbers, we compare them with bootstrapped 95% confidence intervals on individual models, as well as observed variance across different setups. In all our experiments, we consider both the (discrete) metric preferred for the benchmark and a more continuous representation for the same task.

## 3.1 Analysis Methodology

For our initial variance analysis, we use both benchmark-level scores (to compute variance and monotonicity) and sample level scores (to estimate 95% confidence intervals). Here, we provide a brief description of the metrics we compute.

**Seed Mean ($\mu(\mathcal{S}, \mathbb{M})$)**   We compute the performance using metric $\mathcal{S}$ of the final checkpoint (at 210B tokens) of each of the 10 "fully trained" models in $\mathbb{M}$ (one for each seed).

**Seed variance ($\sigma(\mathcal{S}, \mathbb{M})$)**   Given a benchmark, a preferred metric $\mathcal{S}$, and a set of models $\mathbb{M} = \{M_1, M_2, \ldots M_n\}$, we define the benchmark seed variance $\sigma(\mathcal{S}, \mathbb{M})$ as the standard deviation of the metric $\mathcal{S}$ scores $\{\mathbb{S}_{\mathbb{M}} = \mathcal{S}_{M_1}, \mathcal{S}_{M_2} \ldots \mathcal{S}_{M_n}\}$ for each of the models in $\mathbb{M}$.

To estimate the variance expected due only to random seed changes, we take the average of this metric over all checkpoint timesteps $\sigma(\mathcal{S}, \mathbb{M}) = \frac{1}{21} \sum_{time=\{10..210B\}} \sigma(\mathcal{S}, \mathbb{M}^{(time)})$, where for example $\sigma(\mathcal{S}, \mathbb{M}^{(time)})$ corresponds to the standard deviation of performance of the 10 model checkpoints (across seeds) after 200B tokens of training. For each benchmark, we consider both a discrete and a continuous metric.[1] The benchmark and metric details are provided in Table 5 of Appendix A.

**Confidence intervals (95% CI)**   We use the `bootstrapped` library[2] to compute 95% bootstrapped confidence interval (CI) values for each of the benchmarks on all 210 checkpoints from our 10 random seeded pretraining runs. Since bootstrapping is expensive, we also compute analytic interval (for discrete metrics) using the formula:

$$CI_{\texttt{analytic}}(\mathrm{M}) = 1.96 * \sqrt{\frac{\mathcal{S}_{\mathrm{M}} \times (1 - \mathcal{S}_{\mathrm{M}})}{N}},$$

where $\mathcal{S}_{\mathrm{M}}$ is the obtained preferred metric score for model M on a given benchmark and $N$ is the number of test instances present in that benchmark. Empirically, we observe that, for the distributions we consider, bootstrapped and Analytic CIs converge when the number of bootstrap samples is large.

**Monotonicity values (mon_disc / mon_cont)**   We compute the extent to which the scores for a benchmark develop monotonically during training. We define monotonicity for seed $i$ as the Kendall Rank correlation between the list of scores $[\mathcal{S}_{\mathrm{M}_i^{10B}}, \mathcal{S}_{\mathrm{M}_i^{20B}}, \ldots, \mathcal{S}_{\mathrm{M}_i^{210B}}]$ and a monotonically increasing or decreasing array of the same length, for discrete and continuous metrics, respectively.

## 3.2 Results

In this section, we present our comprehensive analysis for two scenarios.

**Seed variance**   In Table 1, we report the observed variance across our 7B seed models in which the training setup is same across all init seeds, including a deterministic data ordering. We contextualise these numbers with the per-model 95% confidence interval, reported in the form of an average of 210 (one for each model) confidence interval sizes. The latter is easily computable from a single training run, whereas the former requires multiple (expensive) training runs with different seeds.

---

[1]With the exception of the datasets Big Bench (Hard), MATH, Natural Questions, and TriviaQA.

[2]https://github.com/facebookarchive/bootstrapped

Table 1: **Variance values on 7B seed models.** Benchmarks are listed in alphabetical order. We report means - $\mu(\mathcal{S}, \mathbb{M})$, standard deviations - $\sigma(\mathcal{S}, \mathbb{M})$, confidence intervals - 95% CI, and monotonicities - $\text{mon}_{\text{disc}}$, $\text{mon}_{\text{cont}}$. We also report size and chance level performance for reference—note all generative tasks have a chance level performance of 0. $\sigma(\mathcal{S}, \mathbb{M})$ is generally lower than 95% CI. We also observe that $\text{mon}_{\text{cont}} > \text{mon}_{\text{disc}}$ for all benchmarks.

| Benchmark | Size | Chance | $\mu(\mathcal{S}, \mathbb{M})$ | $\sigma(\mathcal{S}, \mathbb{M})$ | 95% CI | $\text{mon}_{\text{disc}}$ | $\text{mon}_{\text{cont}}$ |
|---|---|---|---|---|---|---|---|
| AGIEval | 2546 | 20 | 23.44 | 0.77 | 1.63 | 0.37 | 0.29 |
| ARC-C | 1165 | 25 | 39.71 | 0.80 | 2.74 | 0.88 | 0.91 |
| Big Bench (Hard) | 6511 | 0 | 29.10 | 0.87 | 1.07 | 0.77 | - |
| COPA | 100 | 50 | 78.80 | 2.15 | 8.30 | 0.56 | 0.90 |
| GSM8k | 1319 | 0 | 4.10 | 0.41 | 0.87 | 0.74 | 0.30 |
| Hellaswag | 10042 | 25 | 70.08 | 0.21 | 0.93 | 0.99 | 0.99 |
| HumanEval | 164 | 0 | 11.89 | 1.11 | 3.98 | 0.79 | 0.98 |
| MATH | 5000 | 0 | 1.52 | 0.23 | 0.28 | 0.52 | - |
| MMLU | 14042 | 25 | 25.86 | 0.57 | 0.72 | 0.09 | 0.15 |
| MMLU-Cloze | 14042 | 25 | 37.47 | 0.22 | 0.79 | 0.95 | 0.96 |
| Natural Questions | 3610 | 0 | 16.43 | 0.60 | 1.04 | 0.91 | - |
| PIQA | 1838 | 50 | 76.93 | 0.41 | 1.99 | 0.87 | 0.93 |
| SIQA | 1954 | 33 | 46.69 | 0.55 | 2.21 | 0.66 | 0.81 |
| TriviaQA | 11313 | 0 | 42.69 | 0.45 | 0.83 | 0.99 | - |

Table 2: **7B seed models.** Comparison between discrete (Disc) and continuous (Cont) metrics along with the signal to noise ratio (SNR). The means - $\mu(\mathcal{S} = \text{Disc}, \mathbb{M})$, $\mu(\mathcal{S} = \text{Cont}, \mathbb{M})$ and standard deviations (Disc Std, Cont Std) reported here (and used to calculate SNR) are computed across the final checkpoints across the 10 seeds.

| Benchmark | $\mu(\mathcal{S} = \text{Disc}, \mathbb{M})$ | Disc Std | Disc SNR | $\mu(\mathcal{S} = \text{Cont}, \mathbb{M})$ | Cont Std | Cont SNR |
|---|---|---|---|---|---|---|
| AGIEval | 23.44 | 0.93 | 25.20 | 0.2267 | 0.0009 | 254.93 |
| ARC-C | 39.71 | 0.87 | 45.89 | 0.2684 | 0.0007 | 381.64 |
| COPA | 78.80 | 2.04 | 38.63 | 0.5376 | 0.0008 | 662.41 |
| GSM8k | 4.10 | 0.52 | 7.88 | 0.9948 | 0.0653 | 15.24 |
| Hellaswag | 70.08 | 0.12 | 608.23 | 0.2833 | 0.0001 | 1921.15 |
| HumanEval | 11.89 | 1.75 | 6.79 | 0.2186 | 0.0018 | 124.08 |
| MMLU | 25.86 | 0.49 | 52.45 | 0.2511 | 0.0007 | 347.57 |
| MMLU-Cloze | 37.47 | 0.12 | 302.73 | 0.2698 | 0.0004 | 678.42 |
| PIQA | 76.93 | 0.39 | 198.98 | 0.5168 | 0.0003 | 1641.14 |
| SIQA | 46.69 | 0.51 | 91.87 | 0.3656 | 0.0009 | 387.11 |

For some benchmarks (e.g. AGIEval, MMLU), scores are around chance accuracy ($\sim 25\%$) even after training for 210B tokens. Benchmarks with few test examples (like COPA and HumanEval) exhibit high variance (both seed variance and 95% CIs). Generally, the 7B seed variance is well below the 95% CI for the same benchmark, though the ratio of the two is quite variable. Having access to the former value, which is smaller but closer to what would be needed to, for instance, compare two data mixes, may allow practitioners to make more fine-grained decisions during model development.

Motivated by prior work which suggests a move to continuous metrics (Srivastava et al., 2022; Schaeffer et al., 2023; Du et al., 2024; Schaeffer et al., 2024), we show a comparison of discrete and continuous metrics along with their signal to noise ratios (SNR = $\frac{\mu(\mathcal{S}, \mathbb{M}^{210B})}{\sigma(\mathcal{S}, \mathbb{M}^{210B})}$) in Table 2. To maintain consistency, we used probability mass of the predicted answer for all choice-based benchmarks and NLL of the correct answer for generation-based benchmarks; more details are provided in Appendix A. We observe that the SNR is considerably higher for continuous metrics for all benchmarks, suggesting that they may be better when comparing models in the sense that they are less confounded by noise. These results may thus help in building better scaling laws for downstream evaluation tasks (Achiam et al., 2023), along with accurate comparisons between two models that have performances lying within the confidence interval for the discrete metric.

**Monotonicity** In Table 1, we list the monotonicity values for each of the continuous and discrete metrics listed in Table 5. Higher monotonicity values are indicative of evaluations that more stably represent model improvement. In almost all cases, the mononicity is better for the continuous metrics

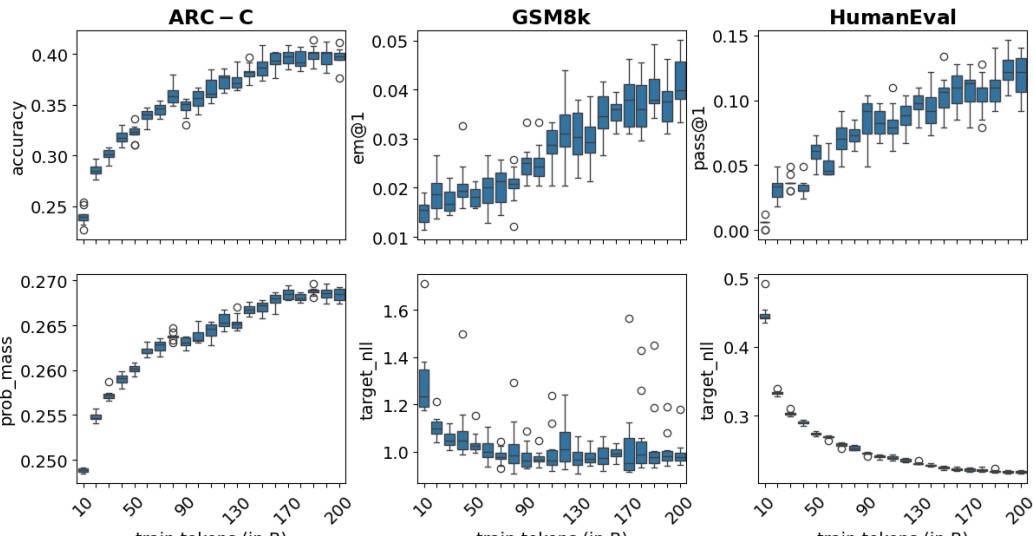

Figure 1: **Development of model performance over time.** Boxplots for both discrete and continous metrics depicting the model improvement over time for ARC-C, GSM8k, and HumanEval. Top row depicts discrete metrics for each of the benchmarks, and the bottom row is composed of the continuous metrics. Continuous metrics develop more stably compared to discrete metrics.

than for the discrete metrics, mirroring our findings with SNR above. However, for some benchmarks, such as HellaSwag and TriviaQA, the difference is minimal, likely since these benchmarks saturate earlier in training. Likewise, for benchmarks where performance remains at chance level we observe very low monotonicities.

In Figure 1, we visualise the development of discrete and continuous metrics and their seed variance during training, for ARC-C, GSM8k, and HumanEval. Generally (with the exception of GSM8k), continuous metrics have better predictive scaling compared to the discrete metrics because they have higher monotonicity and SNR. Interestingly, we see that the variance remains relatively constant as performance increases, suggesting that the estimates may extrapolate well to models trained for longer. Overall, these results suggests that monitoring continuous metrics could be more fruitful during model development than tracking discrete metrics.

### 3.3 THE CURIOUS CASE OF MMLU

Motivated by prior work considering the inconsistency of multiple choice benchmarks (Wang et al., 2024; Alzahrani et al., 2024), we examined two formulations of MMLU: (Standard) MMLU and MMLU-Cloze.

Standard MMLU refers to the prompting format where the choices along with the choice texts are present for the few-shot examples as well as the question in the prompt text. To evaluate the sample, we append the choice letters ("A", "B", "C", or "D") at the end of the prompt text, and pick the choice that has the lowest negative log-likelihood (NLL). For MMLU-Cloze, just the correct choice's text is present for the few-shots, and we pick the choice that gives the lowest NLL after appending the choice texts at the end of the prompt. The prompts used for the two cases are detailed in Appendix B.

In Figure 2, we plot performance over training and see that standard MMLU is at chance performance even after training on 210B tokens. The cloze formulation performs better, and importantly has lower seed variance and much higher monotonicity (0.95 instead of 0.09, see Table 1). This result seems surprising, given that the cloze format is not standard. Further investigation yields that fully-trained large models tend to have better performance on standard MMLU compared to MMLU-cloze (e.g. 78.7% on standard MMLU vs. 60.6% for MMLU-Cloze for LLaMa 3 70B). Despite this difference in absolute performance, we find the performance on standard and cloze formats is highly correlated

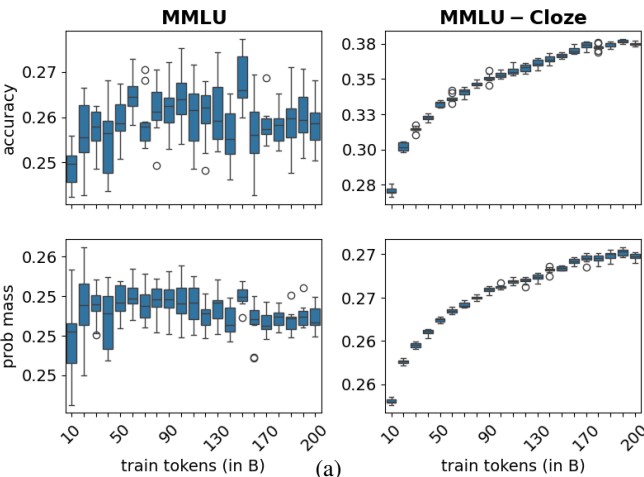

Figure 2: **Development of model performance over time.** In this figure we show the boxplots for the two MMLU variants. The top row is for the discrete metric (accuracy) and bottom row for the continuous metric (probability mass of the correct answer). MMLU-Cloze develops more stably in the earlier stages of pre-training.

for fully trained large models (Pearson correlation of 0.92 on the 70 models listed in § 2). See Appendix C.3 for more ablations on why MMLU-Cloze works better in the initial stages.

Given these results, we encourage researchers to use cloze formulations when doing pre-training, datamix ablations at different compute FLOPs, and building scaling laws, as they are less confounded by noise during early stages of training, but still seem predictive of final performance on the standard MMLU format.

## 4 UNDERSTANDING VARIANCE THROUGH THE LENS OF ITEM ANALYSIS

In the previous section, we computed the empirically occurring variances for commonly used evaluation benchmarks, considering benchmark-level scores, and we showed how looking at continuous metrics or cloze formulations of tasks can boost SNR.

As another avenue of possibly reducing variance, and to better understand it, we take inspiration from *item analysis*, a common method used to assess the usefulness of individual test questions on standardised tests administered to humans (Livingston, 2011; University of Washington, 2024). Item analysis focuses on metrics of individual samples (e.g. difficulty) to understand the types of questions on tests in terms of how individuals (in our case, models) perform on them.

### 4.1 METHOD

In applying item analysis to benchmarks, we consider two metrics. *Item difficulty* refers to the average score on an item across models; *Item discrimination* refers to the correlation between models' performances on a single data point and models' overall performances. Intuitively, items with either high or low difficulty will have low discrimination (as all models will be wrong or right, respectively).

As we wish to make recommendations about evaluation datasets that extend to future models, we split our 70 models into train and test sets. We consider two splits: "random" and "difficulty". As the name suggests, in the random split, we split models randomly; In the difficulty split, we hold out the best performing 14 models. The full lists of models in each split can be found in Appendix D.1. We then calculate item analysis metrics on individual data points for the train and test sets. As is often done with human testing, we also consider the use of removing data points with low item discrimination, and observe the effects this has on evaluation metrics such as mean, standard error of the mean (std. err.),[3] and monotonicity.

---

[3]Note that the confidence intervals of § 3.1 are 1.96 times the standard error.

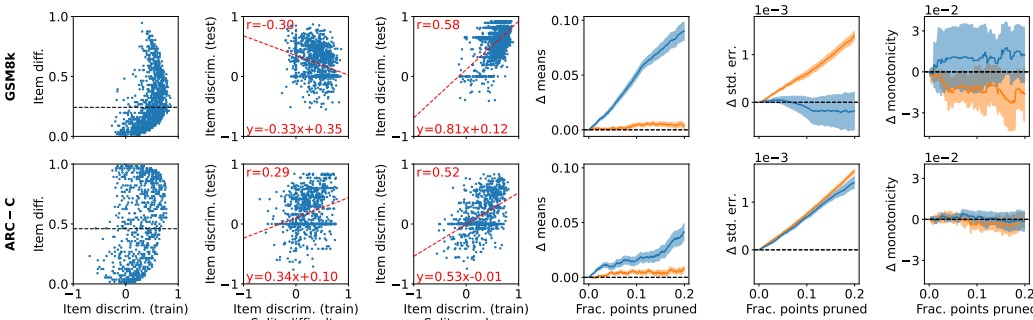

Figure 3: Item analysis results on GSM8k and ARC-C. Results on additional benchmarks provided in Appendix D.2. **First column** shows a scatter plot of item discrimination (x-axis) vs item difficulty (y-axis). **Second column** shows a scatter plot of item discrimination calculated over models from the train or test set of the difficulty split. **Third column** is the same as the second, except on the random split. As expected (since train and test splits come from the same distribution), discrimination on train models for this split is positively correlated to discrimination on test models. **Fourth, fifth, and sixth columns** show the effects of iteratively removing up to 20% of items (based on discrimination) on the mean (fourth column), standard error (fifth column) of model performance on the test set from the difficulty split by looking at the delta. Error bars indicate 95% confidence intervals in the delta. Monotonicity (sixth column) is calculated over the 10 runs from § 2. Orange curves show effects from randomly removing points, as a baseline.

## 4.2 RESULTS

In Figure 3, we show results for two illustrative benchmarks: ARC-C and GSM8k. Full results across other benchmarks can be found in Appendix D.2. Overall, we find that item discrimination scores may not provide much useful signal for the field of language model evaluations (unlike their widespread usage in human standardised testing). This is especially true given that state-of-the-art models perform better and better, and we would like tests to stay informative when models improve. To illustrate this, we show how high discrimination on train (weaker) models often does not correspond to high discrimination on test (stronger) models (Figure 3, second column). Striping around $x = 0$ corresponds to items that train set models always get *wrong* (yielding 0 discrimination) but are informative on test set models. Similarly, striping around $y = 0$ corresponds to items that test models always get *right* (yielding 0 discrimination) but are informative on the train set. If we instead consider item discriminations on a *random* split of models (Figure 3, third column), we see stronger correlations, indicating that the low correlation is in fact due to the difference in item discrimination on weaker and stronger models.

In Appendix D.3, we qualitatively inspect examples with negative item discrimination (which are thus anti-correlated with overall model performance), but are not able to discern any clear patterns for most benchmarks (a notable exception being Hellaswag, see Figure 12). While these negative results suggest item discriminations may not be the most informative means of understanding (or reducing) variance on stronger models, we consider further application to explore the causal effect.

Specifically, we consider pruning data points with low item discrimination, with the hopes that this will reduce variance or improve monotonicity. More precisely, we prune data points with low item discrimination on the train set of models from the difficulty split and we visualise metrics calculated using the pruned subset on the test set of models from the difficulty split. Results are presented in the three rightmost columns of Figure 3. Overall, while we find modest improvements in both standard error (a decrease) and monotonicity (an increase), the drift in the estimated accuracy is mildly concerning. It may be acceptable for the purpose of comparing models, but may also provide an overestimate of capabilities if considering the absolute score. One hypothesis for this discrepancy with human testing could be that item discrimination for human tests typically does not consider out-of-distribution splits – it takes into account the entire spectrum of scores. However, even beyond the difficulty split, we similarly find little-to-no benefits on the random split (see Figure 11). As a result, we overall would not suggest the use of item analysis-based methods for understanding

Table 3: **Variance values for Tiny Benchmark (across seeds).** Full represents the full benchmark, and IRT/IRT++ use the 100 examples proposed in Polo et al. (2024). $\sigma(\mathcal{S}, \mathbb{M})$ is the seed variance defined in § 3.1, which is represented as $\sigma$ in this table.

| Benchmark | Full $\mu$ | IRT $\mu$ | IRT++ $\mu$ | Full $\sigma$ | IRT $\sigma$ | IRT++ $\sigma$ |
|-----------|------------|-----------|-------------|---------------|--------------|----------------|
| ARC-C     | 39.71      | 46.21     | 42.32       | 0.80          | 1.80         | 1.86           |
| GSM8k     | 4.10       | 3.21      | 4.62        | 0.41          | 1.16         | 1.49           |
| Hellaswag | 70.08      | 71.80     | 68.81       | 0.21          | 2.06         | 2.42           |

Table 4: **Monotonicity values for Tiny Benchmark.** We list the monotonicity values for both discrete ($\text{mon}_{\text{disc}}$) and continuous ($\text{mon}_{\text{cont}}$) metrics for the 7B seed models from § 3.2. Full represents the full benchmark, and IRT/IRT++ use the 100 examples proposed in Polo et al. (2024).

| Benchmark | $\text{mon}_{\text{disc}}$ (Full/IRT/IRT++) | $\text{mon}_{\text{cont}}$ (Full/IRT/IRT++) |
|-----------|----------------------------------------------|----------------------------------------------|
| ARC-C     | 0.88 / 0.64 / 0.63                           | 0.91 / 0.78 / 0.82                           |
| GSM8k     | 0.74 / 0.32 / 0.30                           | 0.30 / 0.24 / 0.24                           |
| Hellaswag | 0.99 / 0.84 / 0.80                           | 0.99 / 0.93 / 0.94                           |

variance in language model evaluations, though the underlying cause for this mismatch remains an open question for future work.

## 5 Possible pitfalls of using efficient benchmarking

In a similar category to item analysis, *item response theory* (Cai et al., 2016; van der Linden, 2018; Brzezińska, 2020; Lord & Novick, 1968) describes a set of statistical models used to analyse human abilities on standardised test data. In the recent past, the method has become popular as a means of understanding model performance on a set of evaluation samples (Lalor et al., 2016; Vania et al., 2021; Rodriguez et al., 2021). Most recently, Polo et al. (2024) used IRT to cluster evaluation points with the aims of reducing eval benchmark size (and thus, the cost of running).

Following our mixed findings applying item analysis, we apply the IRT method from Polo et al. (2024) to our models and the overlapping set of evaluation benchmarks. For a brief summary of the IRT method, we refer to Appendix E.1. Specifically, we go beyond the comparisons drawn in prior work and consider how our defined variance metrics (§ 3) change under this model. We believe the application to evaluating intermediate checkpoints during pretraining is especially relevant, as that's the application where smaller evaluation datasets could have the most efficiency gains (as opposed to one-time evaluations of larger models).

In Tables 3 and 4, we report various metrics on the discrete performance measure for GSM8k, Hellaswag, and ARC-C. We find that simply using the performance on the 100 datapoints selected by Polo et al. (2024) for each benchmark can lead to quite large deviations in the mean (an overestimation by 7% for ARC-C). The full IRT++ method obtains less deviation, replicating prior findings (Polo et al., 2024). However, both methods suffer from greatly increased seed variance (final two columns, Table 3), indicating that the tiny-benchmarks method may have limited use during pretraining ablations as it makes model comparisons more likely to be confounded by randomness from the initialisation and data ordering seed. This increased variance is also reflected in the monotonicity metrics – we see a decrease in monotonicity in Table 4, indicating that performance oscillates more during training (see Figure 5). This clearly shows that we cannot use efficient benchmarking techniques for building scaling laws and doing pre-training/datamix ablations.

Beyond the smaller scale models, we also considered the use of tiny-benchmarks for evaluating larger models, like the ones used for item analysis in Section 4. In Figure 4, we find that IRT-based methods generalise relatively well when it comes to the average performance metric (with the IRT++ estimator performing better), but have much larger standard error of the mean. This increased error cautions against the use of IRT-based subsets for model evaluations that will be used to compare different models. To quantify how this increased standard error of the mean may affect model rankings, we also compute the Kendall rank correlation on our 70 models using the performance estimate obtained from using the full dataset, as well as the IRT and IRT++ methods. In Table 7, we find that the

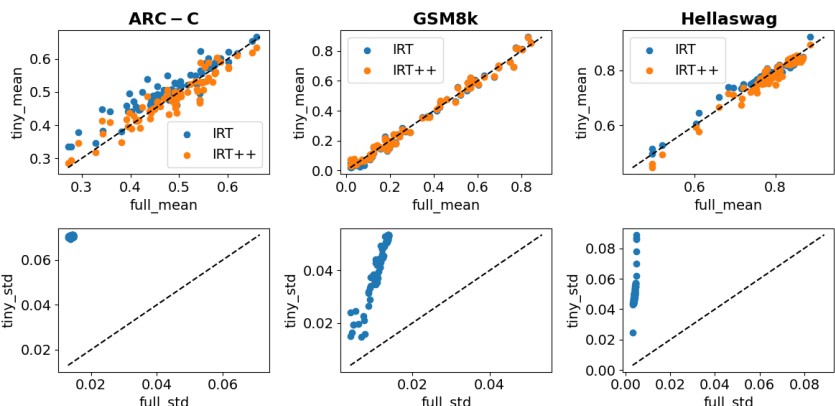

Figure 4: **Tiny Benchmarks Means and Standard Errors of the mean (proportional to 95% CI).**

correlation can drop as low as 0.76, corresponding to 12% of model pairwise comparisons giving the opposite result when using the IRT or IRT++ method (versus the full dataset mean estimate). Furthermore, we find that the number of flips is relatively higher on models that perform better, suggesting that IRT-based methods may not scale well (similar to item analysis). These findings reinforce the promise of IRT-based methods for a point estimate of the mean (relatively low error, Figure 4), but caution against the use of IRT-based methods when *comparing* models due to the increased variance of the estimate. Moreover, the parameters obtained by fitting IRT-based methods on older models do not generalize well to out of distribution models, as clearly seen in Table 3.

## 6 RELATED WORK

While a significant body of work exists proposing natural language processing (NLP) benchmarks to evaluate the capabilities of models, there is comparatively less work studying the benchmarks themselves. Before the era of chat large language models, Marie et al. (2021) conducted a large scale meta-analysis of 769 research papers published from 2010 to 2020 and identified troubling trends, including one that partially motivates our work: models are frequently declared superior to competitors based on small differences in performance scores, without proper hypothesis testing that takes into account natural fluctuations in benchmark scores. Spiritually similar claims were made by Dehghani et al. (2021) in their provocatively titled paper "The Benchmark Lottery". Kocmi et al. (2021) further leveraged large-scale human experiments to evaluate benchmarks with automated metrics and concluded that commonly used metrics such as BLEU score had led to poor deployment decisions. Their conclusion was echoed by a meta-analysis of 3500 NLP benchmark scores published on Papers with Code (Blagec et al., 2022).

More recently, with accelerating progress in NLP, researchers have begun to study benchmarks in earnest to understand their properties and limitations (Gehrmann et al., 2023). Von Werra et al. (2022) proposed a framework to evaluate benchmarks themselves and provided a mechanism for researchers to share their benchmarking analyses. Certain papers have studied specific aspects of benchmarks, focusing on the sensitivity of language models to various factors. Sclar et al. (2023) tested how sensitive language models are to differently formatted prompts, while Wang et al. (2024) and Alzahrani et al. (2024) find that models are inconsistent across changes in the format of MCQA benchmarks. Our work builds on these works by focusing on the inherent variance in benchmarks (e.g. due to model seed) that practicioners should consider when making decisions, and suggesting minor modifications (e.g. in how a task is scored or formulated) that can reduce this variance.

With the aims of improving efficiency in model development cycles, recent work proposes reducing the size of evaluation benchmarks by picking representative samples (Vivek et al., 2023; Polo et al., 2024). Polo et al. (2024) show that methods from human standardised testing (specifically, item response theory; Lord & Novick, 1968) can be combined with clustering to subselect evaluation benchmarks without incurring too much deviation from the mean. However, they do not consider the increased *variance* from their method nor how small deviations in means can compound when comparing multiple models. We go beyond their work by considering the use of additional methods

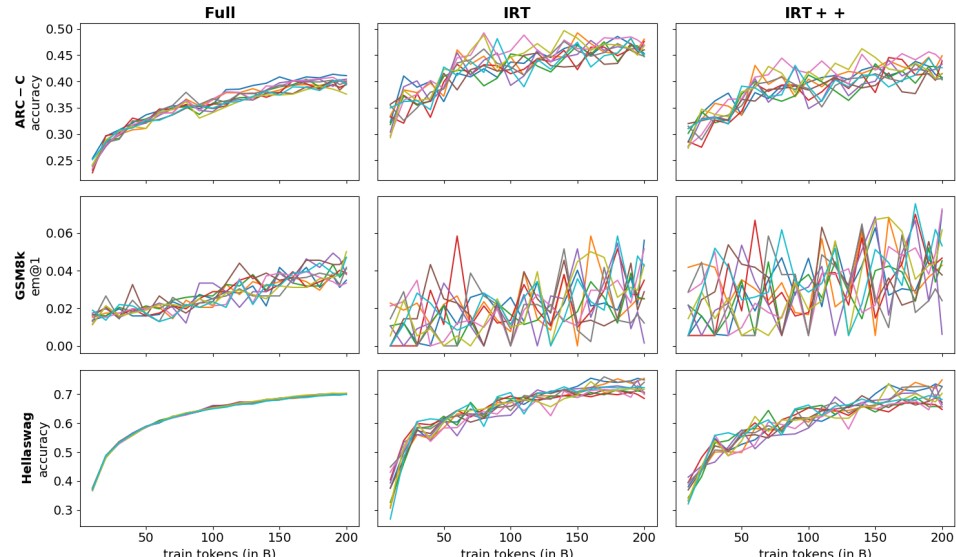

Figure 5: Increased variance when using IRT or IRT++ based estimation of benchmark means during pretraining. While Table 4 shows the decreased monotonicity when estimating with IRT-based methods, here we show performance curves through training for each of the 10 pretraining runs from § 2. Curves are visibly noisier (and less monotonic), showing the increased difficulty pracitioners may have in interpreting results if using IRT-based methods.

from human standardised testing literature (item analysis; Livingston, 2011), as well as showing that such methods generally do not meaningfully reduce variance.

Perhaps most similar to ours is the work of Xiang et al. (2022), who study different sources of variance in NLP benchmarks and offers cautionary advice about when one should (not) be confident in benchmark scores. Their approach is limited to the machine translation setting; here we quantify and study variance in 13 different NLP benchmarks (covering general knowledge, reasoning, coding, and math) across 280 models, including many frontier LLMs.

## 7 CONCLUSION

As language models become more and more prevalent, it has become increasingly important to get a sense of their capabilities. One of the primary ways to assess these capabilities is through the use of evaluation benchmarks, where a model is scored on a series of examples. These scores are often directly compared, without consideration of the *variance*. This obscures the interpretation of evaluation results, in assessing final models as well as making decisions during model development. In this work, we aimed to quantify evaluation benchmark variance across a range of settings (from pretraining intermediate checkpoints, to the largest frontier LLMs) using a diverse set of metrics (seed variance, confidence intervals, and monotonicity). Beyond quantifying variance, we also experimented with various techniques used in human standardised testing (item analysis; University of Washington (2024), item response theory; Cai et al. (2016)), but generally found these methods to be ineffective on the models and benchmarks we considered, in terms of reducing variance. Future work could explore such avenues further, and it is possible that as models reach closer and closer to human-level performance these methods will provide more useful insights. On the other hand, in line with recent work advocating for a *teleological* approach to measuring capabilities (McCoy et al., 2023), we demonstrated LLM-specific techniques (e.g. the use of continuous metrics or cloze-formatted tasks) can improve the signal-to-noise ratio in our evals. Such techniques are not available when assessing humans, but provide a unique opporutnity for LLM evaluations, especially when performing pretraining ablations. We hope our work spurs future work in this direction of reducing variance, in addition to serving as an empirical guide for model practitioners to use when comparing models and assessing performance.

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

# A    MODELS AND BENCHMARKS DETAILS

For pre-training the 7B Llama-2 like checkpoints, we use a pre-training mix of publicly available data. We apply filtering to remove documents containing a high amount of personal information. We use a learning rate of $3.0 \times 10^4$, sequence length of 4096, and a batch size of $4.1M$ tokens to train the 7B models for $50000$ steps. We use 256 80GiB A100 GPUs for a single pre-training run for 50k steps on our internal cluster. We do 10 such runs with different seeds. Each step takes 4.3 seconds.

For running the evaluations, we use 8 GPUs for each evaluation job comprising multiple evaluation datasets in a single job. A single evaluation job takes on average takes 3.5 hours for 13 benchmarks.

In Table 5, we provide the discrete metric (preferred), the continuous metric, and the number of samples for each of the benchmarks we consider. We can choose any continuous metric like character NLL, raw NLL, probability mass, log of probabilities, etc. for the benchmarks, but to maintain consistency, we choose probability mass of the predicted answer for choice-based tasks and negative log likelihood (NLL) of the target answer for generation-based benchmarks. Choice-based benchmarks are evaluated by appending the possible option choice letters or choice texts and then choosing the option with the lowest NLL. Generation-based benchmarks involve free-form generation, where the answer is extracted from the model's response using various post-processing techniques.

For the ARC-C benchmark we exclude 7 problems as 4 of them have only 3 answer choices, and 3 of them have 5 answer choices. We use all the other samples containing 4 choices each.

Table 5: **Benchmark Details** Details of all benchmarks used in the paper alphabetically. Exact Match (EM) is computed for 1 generation (maj@1). Prob Mass is the probability mass of the predicted answer and Target NLL represents the NLL of the target answer. CoT represents chain of thought prompting.

| Benchmark | License | # samples | # few-shot | Disc Metric | Cont Metric |
|---|---|---|---|---|---|
| AGIEval (Zhong et al., 2023) | MIT | 2546 | 3-5 | Acc | Prob Mass |
| ARC-C (Clark et al., 2018) | Apache 2.0 | 1165 | 0 | Acc | Prob Mass |
| Big Bench Hard (Srivastava et al., 2022) | Apache 2.0 | 6511 | 3 (CoT) | EM | - |
| COPA (Roemmele et al., 2011) | BSD 2-Clause | 100 | 0 | Acc | Prob Mass |
| GSM8k (Cobbe et al., 2021) | MIT | 1319 | 8 (CoT) | EM | Target NLL |
| Hellaswag Zellers et al. (2019) | MIT | 10042 | 0 | Acc | Prob Mass |
| HumanEval (Chen et al., 2021) | MIT | 164 | 0 | Pass@1 | Target NLL |
| MATH (Hendrycks et al., 2021) | MIT | 5000 | 4 (CoT) | EM | - |
| MMLU (Hendrycks et al., 2020) | MIT | 14042 | 5 | Acc | Prob Mass |
| Natural Questions (Kwiatkowski et al., 2019) | MIT | 3610 | 5 | EM | - |
| PIQA (Bisk et al., 2020) | Academic Free | 1838 | 0 | Acc | Prob Mass |
| SIQA Sap et al. (2019) | - | 1954 | 0 | Acc | Prob Mass |
| TriviaQA Joshi et al. (2017) | Apache 2.0 | 11313 | 5 | EM | - |

Table 6: **Model Details** Details of all models in the paper categorized by model family along with the number of parameters.

| Model Family | Models | Model Sizes (# params) |
|---|---|---|
| Meta-Llama (AI@Meta, 2024; Touvron et al., 2023b;a) | Llama-1, Llama-2, Llama-3 | 7-70B |
| Google (Mesnard et al., 2024) | Gemma | 2-7B |
| Databricks (Databricks, 2024) | DBRX-Base | 132B |
| Mistral (Jiang et al., 2023; 2024) | Mistral, Mixtral | 7-141B |
| Qwen (Bai et al., 2023) | Qwen-1.5 | 0.5-110B |
| EleutherAI (Biderman et al., 2023) | Pythia | 1-12B |
| TII-UAE (Almazrouei et al., 2023) | Falcon | 7-40B |
| DeepSeek (Bi et al., 2024; DeepSeek-AI, 2024) | DeepSeek, DeepSeek-MoE, DeepSeek-V2 | 7-236B |
| StabilityAI (StabilityAI, 2024) | StableLM | 1.6-7B |
| MosaicML (MosaicML NLP Team, 2023) | MPT | 7-30B |

## B  MMLU PROMPT FORMATS

We use the following prompt variations for the standard and cloze versions of MMLU. We list down the preamble and the shot formatting for both cases. The final question is formatted like the few shot examples without the gold choice letter or text.

### B.1  MMLU

> **Preamble**:
> The following are multiple choice questions (with answers) about <*subject*>.
>
> **Shot formatting**:
> <*question*>
> A. <*choice A text*>
> B. <*choice B text*>
> C. <*choice C text*>
> D. <*choice D text*>
> Answer: <*gold choice letter*>

### B.2  MMLU-CLOZE

> **Preamble**:
>
> **Shot formatting**:
> <*question*>
> Answer: <*gold choice text*>

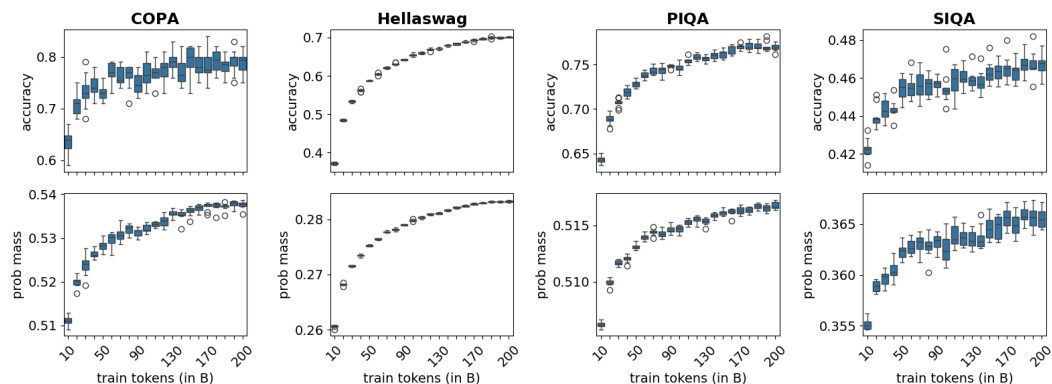

Figure 6: **Development of model performance over time.** Boxplots for both discrete and continous metrics depicting the model improvement over time for COPA, Hellaswag, PIQA, and SIQA. Top row depicts discrete metrics for each of the benchmarks, and the bottom row is composed of the continuous metrics.

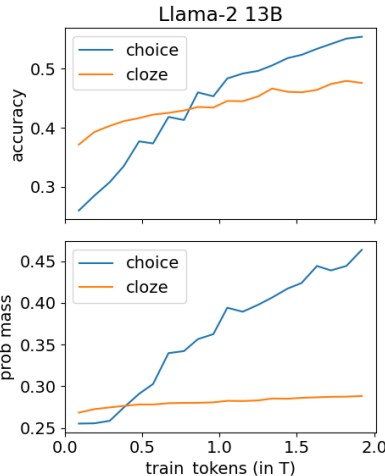

Figure 7: In this figure we show the comparison of the standard (choice) and cloze variants on a Llama-2 13B model trained from scratch.

## C    VARIANCE ANALYSIS ADDITIONAL RESULTS

### C.1    MORE BENCHMARKS

In this section, we present additional results on model performance development for the remaining benchmarks - COPA, Hellaswag, PIQA, and SIQA (see Figure 6). This supplements the results presented in Figure 1 and Figure 2. We observe similar trends except for SIQA. The error bars for both discrete and continuous metrics are similar, however, the continuous metric plot has less number of outliers.

### C.2    ABLATIONS ON CONTINUOUS METRICS

We also show the comparison of model performance development for two different continuous metrics, char-length normalized NLL and token-length normalized NLL for GSM8k and HumanEval in Figure 8.

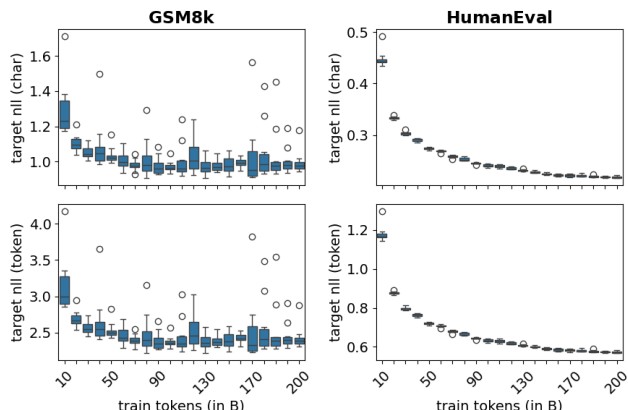

Figure 8: In this figure we show the comparison of the two different continuous metrics (character-vs token-length normalized NLL).

## C.3 MMLU Ablations

To understand why MMLU-Cloze works better in the earlier stages (§ 3.3) whereas the final MMLU performance is higher during the later stages, we train a Llama-2-13B-like model from scratch on our pre-training mix. We observe a sudden jump in performance at around 800B tokens (for both discrete and continuous metrics), after which standard MMLU performs better than MMLU-cloze (see Figure 7).

## D    ITEM ANALYSIS ADDITIONAL RESULTS

### D.1    SPLITS

We used 70 base models for the item analysis results. We provide the splits used below.

**Difficulty split (train):** LLaMa 3 8B, Mistral 7B, Qwen {0.5, 1.8, 4}B, LLaMa 2 7B, LLaMa 2 13B, LLaMa 2 70B, DeepSeek 7B, DeepSeek MoE 16B, Falcon 7B, Falcon 40B, Gemma 2B, Gemma 7B, LLaMa 1 {7, 13, 33, 65} B, MPT 30B, Pythia {1, 1.4, 2.8, 6.9, 12}B, StableLM {3, 7}B. In addition to these open source models, we use 30 internal checkpoints from LLaMa-architecture models we pre-trained on our interal data mix.

**Difficulty split (test):** LLaMa 3 70B, Mixtral 8x{7,22}B, Qwen 1.5 {7, 13, 32, 72, 110}B, DBRX, DeepSeek 67B, and 4 internal held out models.

**Random split (train):** LLaMa 3 {8, 70}B, Mistral 7B, Mixtral 8x{7,22}B, Qwen 1.5 {0.5, 1.8, 4, 7, 13, 32, 72}B, LLaMa 2 7B, LLaMa 2 13B, LLaMa 2 70B, DBRX, DeepSeek MoE 16B, DeepSeek 67B, Falcon 40B, Gemma 2B, Gemma 7B, LLaMa 1 {7, 33, 65} B, MPT 30B, Pythia {1, 1.4, 2.8, 6.9, 12}B, StableLM 3B. In addition to these open source models, we use 25 internal checkpoints from LLaMa-architecture models we pre-trained on our interal data mix.

**Random split (test):** DeepSeek 7B, Falcon 7B, Qwen 1.5 110B, LLaMa 1 13B, StableLM 7B, and 9 internal checkpoints.

### D.2    ADDITIONAL RESULTS

We present results on additional benchmarks, in a similar format to Figure 3, in Figure 9. Furthermore, we provide extended results on the random split of models in Figure 10 and Figure 11.

We provide the stability of the results of random splits across five different sets of train and test models. Figure 10 shows the scatter plots of item discrimination over the train and test set of models. This shows that the results across random splits are robust, and low correlation in the train and test sets for the difficulty split are because of the differences in model capability across the two sets.

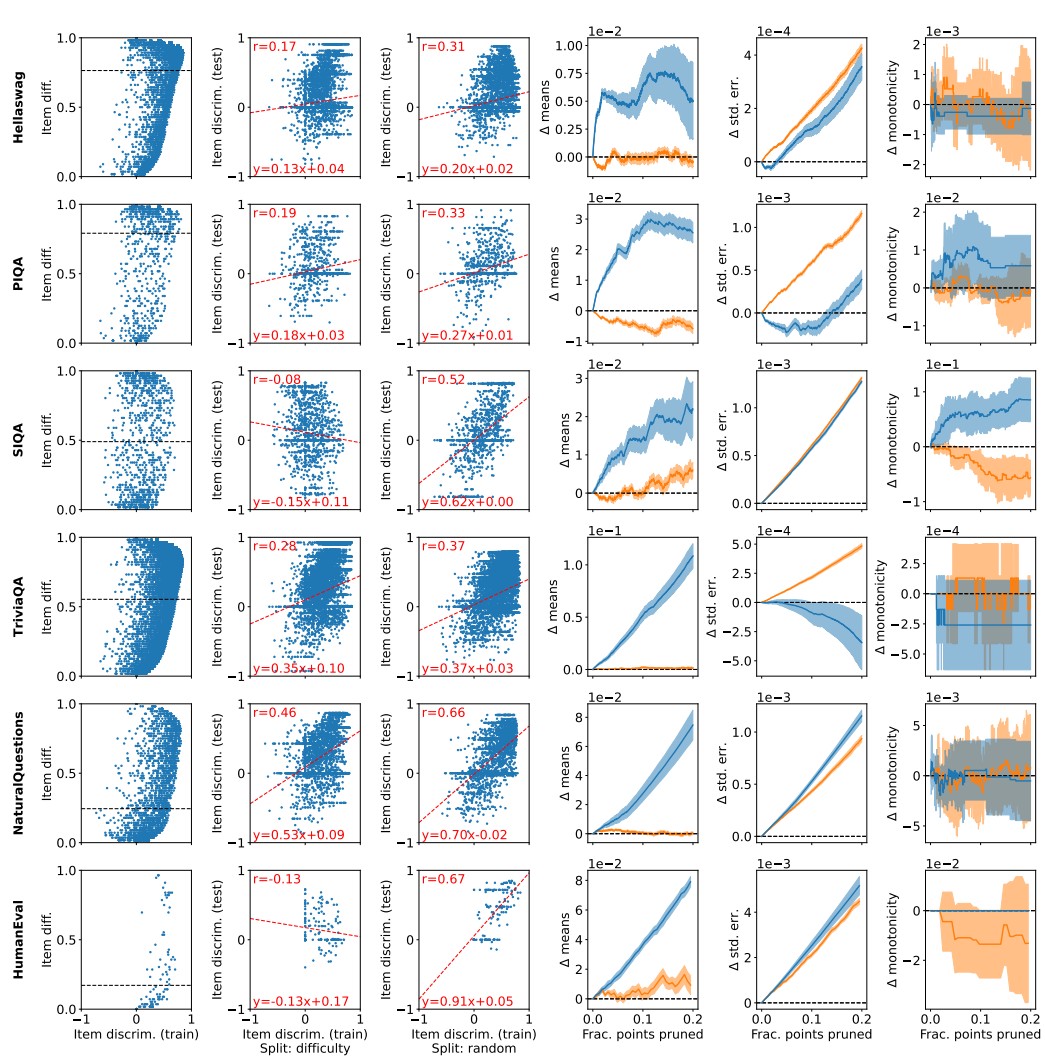

Figure 9: Item analysis results on six additional benchmarks, in the same format as Figure 3.

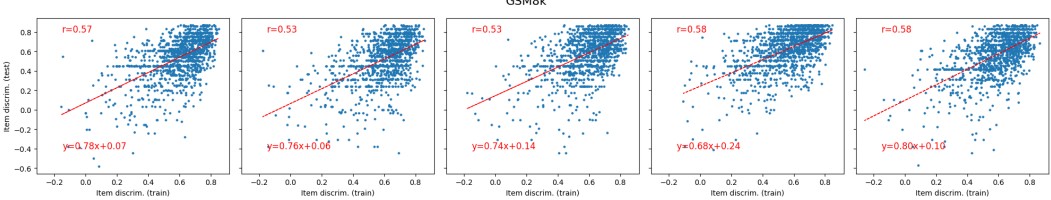

Figure 10: Item discrimation results on five different train/test splits.

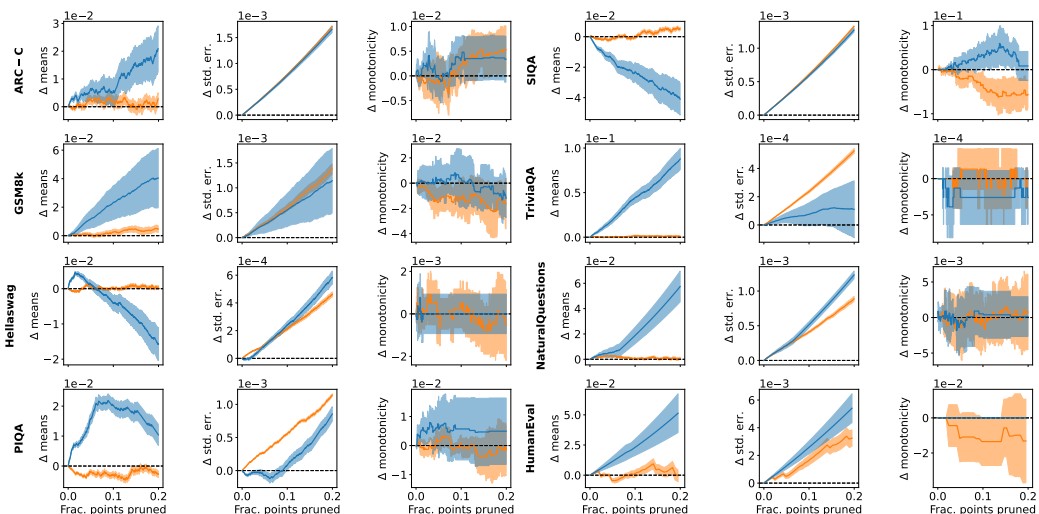

Figure 11: Results on 8 benchmarks when removing points based on item discrimination on the *random* split. These plots are similar to the final 3 columns in Figure 3 and Figure 9. Specifically, we show the effects of iteratively removing up to 20% of items (based on discrimination) on the mean (first column), standard error (second column) of model performance on the test set from the random split by looking at the delta. Error bars indicate 95% confidence intervals in the delta. Monotonicity (sixth column) is calculated over the 10 runs from Section 2. Orange curves show effects from randomly removing points, as a baseline. As we can see, these plots look qualitatively similar to Figure 3 and Figure 9 indicating that the observed lack of benefit from pruning based on item discrimination is not simply due to using the *difficulty* split of models.

### D.3 INSPECTION OF SAMPLES WITH LOW ITEM DISCRIMINATION

We provide the 3 items from GSM8k, ARC-C and Hellaswag with the lowest item discrimination.

**For GSM8k:**

---

**Question**:
Aaron and Vanessa were relay race partners on a running team. Aaron was able to run each mile twice as fast as Vanessa, but Vanessa was able to run twice as far as Aaron did. If Vanessa ran 4 miles and Aaron completed his part of the race in 16 minutes, how long in minutes did Vanessa take to complete her part?

**Answer**: 64

**Item Discrimination**: -0.264

**Item Difficulty**: 0.1

---

**Question**:
Suzie loves to chew fruit-flavored gum. She bought four packs of gum the last time she was at the store. She got two packs of her favorite flavor, strawberry. She paid $2 for a pack of grape gum that she also liked. She wanted to try something new, so she paid half as much for a small pack of green apple gum. If she paid $7 in all, how many dollars did each pack of strawberry gum cost?

**Answer**: 2

**Item Discrimination**: -0.198

**Item Difficulty**: 0.229

---

**Question**:
John brings his dog to the vet. His dog needs 2 vaccines, which are $20 each, and a heartworm check. The heartworm check is 60% of his total bill. If he brought $125 with him, how much does he leave with?

**Answer**: 25

**Item Discrimination**: -0.196

**Item Difficulty**: 0.057

---

**For ARC-challenge (correct answer is italicized):**

---

**Question**:
Wolves, which are top predators, were eliminated from Yellowstone National Park in the 1930s. In 1995, wolves were reintroduced into Yellowstone. During the period in which wolves were absent from Yellowstone, which most likely occurred?

*A. an increase in competition for food resources among small prey*
B. a greater opportunity for primary producers to flourish
C. an increase in the population of tertiary consumers
D. a greater balance of predator-prey relationships

**Item Discrimination**: -0.689

**Item Difficulty**: 0.2

---

**Question**:
Which of these traits is inherited but greatly influenced by the environment?

A. tongue rolling ability
*B. athletic performance*
C. language
D. color of eyes

**Item Discrimination**: -0.574

**Item Difficulty**: 0.443

**Question**:
Organisms interact in the flow of energy in an ecosystem. Carnivores and omnivores are classified as consumers. Which two organisms are also classified as consumers?

A. bacteria and fungi
B. fungi and scavengers
*C. parasites and herbivores*
D. decomposers and herbivores

**Item Discrimination**: -0.539

**Item Difficulty**: 0.071

**For Hellaswag:**

**Question**:
The sunburned man is taking his shirt off and laying it on the bed. His friends help him with cream on his sunburn. the woman

A. places orange-colored tissue paper onto the sunburn.
B. is helping him putting on sunscreen.
C. is getting massage by a man.
*D. is sitting at the table eating.*

**Item Discrimination**: -0.637

**Item Difficulty**: 0.057

**Question**:
A person is seen playing an accordion on a busy street while many people walk around him and watch. the man

A. continue playing with others in the street and ends with him walking away.
*B. continues to play the instrument and ends by stopping to laugh and smile at others.*
C. continues to play behind a set of drums while people walk in and out of frame.
D. continues to play while looking out at people and pans back to the camera.

**Item Discrimination**: -0.551

**Item Difficulty**: 0.086

**Question**:
A female weight lifter bends at the knees. She lifts a barbell to her chest. she

*A. then lifts it over her head before dropping it heavily to the ground.*
B. lowers the barbell and stands, then sways.
C. lifts it over her head.
D. then lifts it over her head to her body.

**Item Discrimination**: -0.488

**Item Difficulty**: 0.229

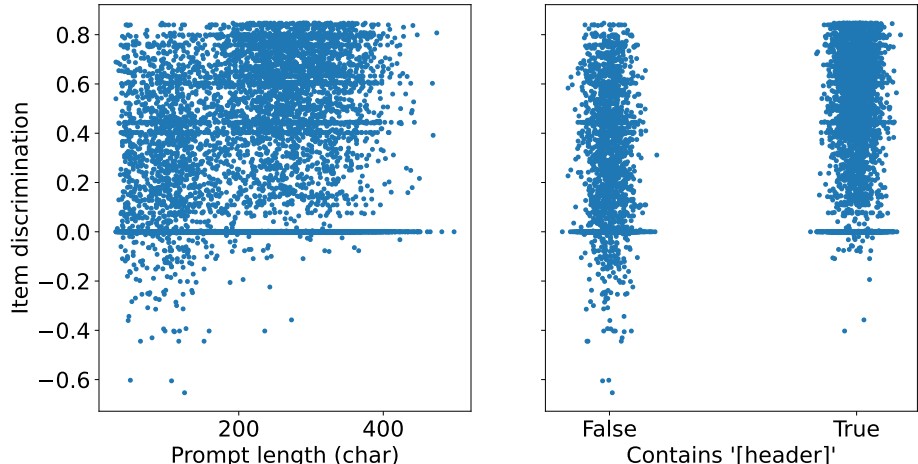

Figure 12: Scatter plots of two features correlated with item discrimination (calculated on the train set of models from the difficulty split). Low item discrimination tends to correspond to short prompts that do not contain '[header]' tags.

Note that for Hellaswag, we did find some correlations to item discrimination in terms of features of the problems. Specifically, as shown in Figure 12, we found that items with low discrimination tend to feature shorter prompts and do not contain tags such as '[header]' in the prompt.

## E    ITEM RESPONSE THEORY ADDITIONAL INFORMATION

### E.1    A BRIEF PRIMER ON IRT

While IRT can refer to a variety of methods, here we focus on the two-parameter multidimensional IRT model used by Polo et al. (2024) to make tiny-benchmarks. Specifically, we define a matrix of model scores on a set of evaluation examples, $Y$, such that $Y_{ms}$ is the score of model $m$ on evaluation example $s$. As this model is mostly applied to discrete metrics in our cases (e.g., accuracy), we focus our exposition on the case where $Y_{ms} \in [0, 1]$ (see Polo et al. (2024) for details on extending to continuous metrics). The IRT model then learns vector embeddings for each model, $\theta_m$, vector embeddings for each example $\alpha_s$ as well as a scalar bias for each example $\beta_s$ to maximize the likelihood of the observations:

$$ P(Y_{ms} = 1 | \theta_m, \alpha_s, \beta_s) = \frac{1}{-\alpha_s^\top \theta_m + \beta_s} $$

Polo et al. (2024) then learn values of $\theta_m, \alpha_s, \beta_s$ for a set of train models across a range of benchmarks. Then, they perform clustering on the evaluation samples where the embedding of each sample is given by $(\alpha_s, \beta_s)$. Finally, they subselect 100 data points that are the most representative and assign weights equal to the size of their clusters.

For a new model, they propose two methods for evaluation. In the first, which is termed "IRT" (to match their paper), we simply use the weighted performance of a model on the 100 data points they identify. In the second, which is termed "IRT++", we consider a weighted combination of "IRT" and an adjusted estimate (which is achieved by 1. learning a $\theta_m$ for the new model on the 100 evaluated data points, using fixed $\alpha_s, \beta_s$, then 2. using the learned $\theta_m$ with the fixed $\alpha_s, \beta_s$ for *all* data points to estimate model performance). Polo et al. (2024) find IRT++ to outperform the IRT estimator, which we reproduce (see Figure 4).

For a full description of the method, we refer the reader to Polo et al. (2024)—we include this primer here for completeness.

Table 7: **Change in model ranking when using IRT-based methods** We compute and list the Kendall rank correlation $\tau$ between model ordering when using IRT-based estimates for each benchmark. To contextualize these, we also compute the percentage of pairwise comparisons which would be flipped (denoted **%**). We also show results limited to the 14 best performing models (the test set of the difficulty split—see Appendix D.1) in the last two columns.

| Benchmark | IRT $\tau$ | IRT++ $\tau$ | IRT % | IRT++ % | IRT % (diff.) | IRT++ % (diff.) |
|---|---|---|---|---|---|---|
| ARC-C | 0.759 | 0.798 | 12.17 | 10.09 | 4.40 | 5.49 |
| GSM8k | 0.913 | 0.912 | 4.51 | 4.51 | 10.99 | 10.99 |
| Hellaswag | 0.881 | 0.794 | 5.96 | 10.35 | 15.38 | 30.77 |

## E.2 ADDITIONAL RESULTS USING TINYBENCHMARKS

See Table 7.

