# OpenReview forum: "Quantifying Variance in Evaluation Benchmarks"
_ICLR.cc/2025/Conference — Submitted to ICLR 2025_

### Official Review · Reviewer_w3M8 · 2024-10-31

**Soundness:** 3
**Presentation:** 2
**Contribution:** 2
**Rating:** 3
**Confidence:** 4

**Summary:**

Authors address the question of measuring variance in standardized evaluation benchmarks. They do so by constructing various ways to assess variance including variance due to different init seeds and looking at evaluation monotonicity at checkpoints in course of pre-training. Authors study a number of already pre-trained models and also train Llama like models from scratch on 7B scales, varying their init seed and obtaining intermediate checkpoints to use for variance estimation. Authors show evidence for continuous metrics delivering better signal-to-noise ratio across all benchmarks, also showing better monotonicity for continuous metrics than for discrete ones. Authors also look into techniques used in human performance testing like item analysis and item response theory and find those not useful for assessing model performance and improving SNR.

**Strengths:**

Question of which metrics used for benchmarking model capabilities are properly reflecting model performance and allow differentiation of models already in early training stage or on smaller scales in general, where performance signal is weak, is very important, as it also leads to better methods for scaling law derivation and proper model comparison via derived scaling laws.

**Weaknesses:**

- Claims put forward by authors are made by experimenting with base models only. Especially when taking benchmarks that have certain question formatting at their core - MMLU, ARC, HellaSwag - it seems to me a strong confound when attempting to make a statement about ability of a benchmark to measure model capabilities. Base models do not handle instruction format well, in general struggling with handling question-answer interaction. This is usually installed via instruction tuning, after which models adhere to various instruction like interactions. I think thus to make proper statements about effect of variance or in general about how to render benchmark signal useful for predictions about model quality, benchmarks that do not rely on certain instruction template are not a good choice when dealing with base models. In my opinion, when working with such benchmarks, authors should have instruction-tuned base models, conducting measurements after that. Benefit of continuous metrics might be thus due to base models not handling instruction like benchs and not due to actual benchmark content related to complexity of problems posed in it. Same might hold for the "curious MMLU" case presented by authors, where cloze variant shows smaller variance and better SNR than multiple choice format standard MMLU form - this might be again just due to base models not being able to handle problem formulation template.

- Further weakness in my opinion is that many of the claims are based on pre-training from scratch done only on one model scale of 7B. It might be more insightful to see trends across scales, even smaller ones. Experiments on larger scale are though expensive, even if going bit further to 13B.

- It is also not quite clear how the claim of better suitability of continuous metrics is backed up. It seems to me there is no clear evidence presented by authors that using those metrics eg indeed allows better prediction from earlier to later training stages or from smaller to larger scale pretraining.

**Questions:**

Can authors show similar evidence for instruction tuned models? It seems to me that it is not expensive to instruction tune base models authors already have obtained, or at least use already available instruction tuned models. Alternatively, can authors present more benchs that do not heavily rely on instruction like templates?

Can authors present evidence that using continuous metrics they can do better on predicting either end model performance from early training phases or larger scale model performance from smaller scale pretraining?

---

> ### Author Response · Authors · 2024-11-25
> **Response to Reviewer w3M8**
>
> We thank the reviewer for their time and address their feedback/questions below:
>
> > Base models do not handle instruction format well, in general struggling with handling question-answer interaction. This is usually installed via instruction tuning, after which models adhere to various instruction like interactions. I think thus to make proper statements about effect of variance or in general about how to render benchmark signal useful for predictions about model quality, benchmarks that do not rely on certain instruction template are not a good choice when dealing with base models.
>
> We would like to point out that instruction-tuning has little effect when conducting evaluations using log-likelihood (https://huggingface.co/blog/open-llm-leaderboard-mmlu). In this setup, the possible choices/text completions are appended to the prefix prompt and the option with the lowest NLL is chosen as the model response. In our paper, this corresponds to AGIEval, ARC-C, COPA, Hellaswag, MMLU, PIQA, and SIQA. Moreover, Dubey et. al, https://arxiv.org/abs/2407.21783 (Figure 14) shows that the prompt variations and option orders are fairly robust and exhibit negligible variance in this setup of NLL-based evaluation. This is a standard setup for evaluating pre-trained base models as seen in GPT series (GPT-3: https://arxiv.org/abs/2005.14165, GPT-4: https://arxiv.org/abs/2303.08774), Llama series (Llama 1: https://arxiv.org/abs/2302.13971, Llama 2: https://arxiv.org/abs/2307.09288, and Llama-3 https://arxiv.org/abs/2407.21783) and other model releases like Mistral/Gemini as well.
>
> > In my opinion, when working with such benchmarks, authors should have instruction-tuned base models, conducting measurements after that. Benefit of continuous metrics might be thus due to base models not handling instruction like benchs and not due to actual benchmark content related to complexity of problems posed in it.
>
> Since we study the variance that is useful for building scaling laws/doing pre-training datamix ablations, it doesn’t make sense to do instruction fine-tuning for each FLOPs scale and at every pre-trained checkpoint. Instruction-tuning should only be done at the end and their evaluation setup is different from those of pre-trained models. Pre-trained evaluations are essential during the early stages of pre-training for selecting the best architecture/datamix that best fits the scaling law curves.
>
> > Same might hold for the "curious MMLU" case presented by authors, where cloze variant shows smaller variance and better SNR than multiple choice format standard MMLU form - this might be again just due to base models not being able to handle problem formulation template.
>
> Again, we would like to emphasize that we are performing evaluations using log-likelihood (details presented in Appendix A), in which the model is not doing any generations, and we limit the output space to the possible option letters/texts, and compute NLLs and choose the option with the lowest NLL as the model’s prediction.
>
> > Further weakness in my opinion is that many of the claims are based on pre-training from scratch done only on one model scale of 7B. It might be more insightful to see trends across scales, even smaller ones. Experiments on larger scale are though expensive, even if going bit further to 13B.
>
> We agree with the reviewer, however, we would like to point out that for building scaling laws, the FLOPs budget matters more as opposed to different model scales. We train 7B models for 210B tokens accounting for a FLOPs budget of $10^{22}$, which is a fairly standard budget for building scaling laws. For example, Dubey et al., 2024 (Llama 3: https://arxiv.org/pdf/2407.21783) and Hoffman et al., 2022 (Chinchilla: https://arxiv.org/pdf/2203.15556). We believe that the reference variance values are representative of this budget: Table 2 column 3 - Disc Std for $10^{22}$ budget, and Table 1 column 5 - $\sigma(\mathcal{S}, \mathbb{M})$ for all budgets $\leq 10^{22}$.
>
> > It is also not quite clear how the claim of better suitability of continuous metrics is backed up. It seems to me there is no clear evidence presented by authors that using those metrics eg indeed allows better prediction from earlier to later training stages or from smaller to larger scale pretraining.
>
> All of the results in Sections 3.2 and 3.3 point to the utility of continuous metrics. Table 2 shows how continuous metrics have higher SNR compared to discrete metrics. Figures 1, 2, and 6 show how continuous metrics are better for predictability and stability in tracking as the seed variance is lower (represented by the box heights in the plots).

---

> > ### Author Response · Authors · 2024-11-25
> > **Response to Reviewer w3M8 (continued)**
> >
> > > Can authors show similar evidence for instruction tuned models? It seems to me that it is not expensive to instruction tune base models authors already have obtained, or at least use already available instruction tuned models. Alternatively, can authors present more benchs that do not heavily rely on instruction like templates?
> >
> > > Can authors present evidence that using continuous metrics they can do better on predicting either end model performance from early training phases or larger scale model performance from smaller scale pretraining?
> >
> > We believe that we have answered both the questions raised by the reviewer above as part of addressing the weaknesses. We are happy to discuss any additional questions that the reviewer may have.

---

> ### Comment · Reviewer_w3M8 · 2024-11-30
>
> > Again, we would like to emphasize that we are performing evaluations using log-likelihood (details presented in Appendix A), in which the model is not doing any generations, and we limit the output space to the possible option letters/texts, and compute NLLs and choose the option with the lowest NLL as the model’s prediction.
>
> Thanks for pointing out the mode of evals being throughout NLL. However, my point was that, independent of output eval, form of the problem formulation itself presented on the input is instruction template based (for standard MMLU), and therefore base model processing input in such format would struggle especially in early training phases, in contrast to MMLU cloze that is not relying on such formatting (as also described in Appendix B). I think that might be a confound when stating difference between MMLU standard and MMLU cloze that is apart from discrete vs continuous metrics. I guess one way to test it would be to have a discrete version that is not relying so heavily on specific instruction format to pose the problem on the input. I think same can be argued in general - to show difference discrete vs continuous, one should remove as far as possible confound of presenting problems in a instruction template form for both evals if testing with base models.
>
> >  ... for building scaling laws, the FLOPs budget matters more as opposed to different model scales ...
>
> I cannot quite follow this point. While FLOP budget surely matters, we would like to extrapolate observed trends towards larger scales. This can be only done well if scanning through a scale span broad enough on smaller scales. I cant imagine how we can do it from few or even single point (no matter whether FLOP related, or combined model/data scale). I think it would be important to observe how variance behaves on a span up to some reasonable high FLOP value, including scan through model scales, as it might be that variance also behaves differently depending on model scale, and this can be overlooked if estimating from single model scale only. This is also in line with comment from other reviewers, eg 7Zuc https://openreview.net/forum?id=E2RyjrBMVZ&noteId=C89i4E8Bo3  I think it would be important to see whether on smaller model scales, that are of course also important for deriving scaling laws in practice, same advantage continuous vs discrete metrics can be stated. This might also allow a bonus to predict some properties of variance for higher FLOP/model scales without running experiments there, although I agree that 7B is reasonable upper threshold here.  Smaller scales <7B are not so expensive, so not sure why would not it be a good thing to do.
>
> In general, I see the merits of the study in attempt to clearly point out advantages of continuous metrics. However, I still find the write up hard to read. I also still struggle to grasp the relevancy of the item analysis and item response theory (IRT) part for the presented results.  The work seems to be building up on https://arxiv.org/abs/2406.04391, and I wish for the same level of clarity achieved there to give a better score that is still not there.

---

### Official Review · Reviewer_VsPx · 2024-11-03

**Soundness:** 1
**Presentation:** 2
**Contribution:** 3
**Rating:** 3
**Confidence:** 2

**Summary:**

This paper quantifies and highlights the important issue of seed variance, amongst different sources of variance, in evaluating language model performance *during pretraining*, emphasizing how insufficient variance quantification in benchmarks can obscure statistically significant performance differences within the same pretraining run.

**Methodology**: It retrains 10 LLaMA 7B models from scratch, and then studies the seed variance along with different metrics throughout the pretraining runs, such as-- 95% confidence intervals, and monotonicity across diverse benchmarks, offering a reference for understanding performance variation across setups.

**Claim 1**: Continuous metrics exhibit higher monotonicity and lower variance than discrete metrics and should be used more widely.

*Application*: Continuous metrics and cloze formats show lower variance and higher signal-to-noise ratios compared to traditional discrete measures, especially in smaller LLaMA 7B models. Simple modifications, such as reframing choice tasks as completion tasks, appear promising for reducing variance.


**Claim 2**: Efficient benchmarking methods can inadvertently increase variance, and it’s essential to verify if methods are distinguishable before using them.

*Application*: Techniques from standardized testing, such as item analysis and item response theory, are found ineffective in meaningfully reducing variance for large language models.

**Strengths:**

Key takeaways I appreciated:

- Continuous metrics show higher monotonicity and lower variance than discrete metrics, making them preferable. I really like this point!
- Variance (likely from benchmarking, not seed variance) could obscure the effectiveness of pretraining by making it harder to track performance improvements.

Benchmarking variance is a promising and underexplored area, especially given the recent focus on reasoning and the close performance of current models on benchmarks-- and seed variance can have a lot of effect.

**Weaknesses:**

See Weaknesses in the order of importance. If there is lack of time, please prioritize the earlier questions:

**P1: Seed Variance and Its Impact on Benchmark Scores**

> We provide a comprehensive reference guide for what magnitudes of variance are expected for what benchmarks across various circumstances

I worry the paper title/abstract/contribution1 is overclaiming as the evidence does not support the above claim-- It seems to me that paper focuses only on a single source of variance: seed variance. Given that studying variance is a common practice (the most popular benchmark, Chatbot Arena provides variance along with scores: https://lmsys.org/blog/2024-04-19-arena-hard/) and previous research on variance sources such as benchmarking details [1] and prompt design [2], the question studied here becomes: Does seed variance contribute overwhelmingly to the overall variance which makes it a critical factor to study, beyond [1,2,Arena]?

I would appreciate:

a) A comparative analysis showing the relative impact of seed variance versus other variance sources on benchmark results.

b) A better distinction in the work for why studying seed variance specifically is important, given the existing literature on other variance sources.

Since the authors claim:
>  if we cannot ‘trust’ our evaluation results or do not understand what improvements are statistically significant, we cannot make sound comparisons, thus making it more challenging to reliably use benchmarks during model development.

a) I could not see the results of the 32 models apart from Llama-2-7B seed runs anywhere. Would be great to know whether the variance obtained Llama-7B seed runs can demonstrate that improvements reported in those models being deemed not-significant on some benchmarks.
b) To what degree are these variance estimates transferable beyond the Llama-2-7B model?.

**Overall:** Currently, it seems to me that, variances from [1] and [2] are critical but easy-to-obtain, while seed variance is compute-heavy to obtain and potentially not as significant source given different prompts and small benchmark details can cause far larger performance shifts. I request authors to present evidence or arguments suggesting this is not the case, and I am open to changing my mind.

[1] When Benchmarks are Targets: Revealing the Sensitivity of Large Language Model Leaderboards

[2] Quantifying language models’ sensitivity to spurious features in prompt design or: How I learned to start worrying about prompt formatting.

**P2: Efficient Benchmarking in High-Variance Contexts**

(a) The rationale here is unclear.
Q1. If variance preservation is a priority, why subsample to an extreme degree of 100 samples? What are the trade-offs between efficiency and variance preservation in benchmarking?

Q2. Conversely, does aggregating several 100 sample estimates reduce variance while still providing efficiency gains compared to evaluating on the whole dataset? Or in most concrete scenarios, efficient benchmarking just might be limited?

(b) The paper says:
> While these negative results suggest item discriminations may not be the most informative means of understanding (or reducing) variance on stronger models

> we overall would not suggest the use of item analysis-based methods for understanding variance in language model evaluations

I doubt those papers claim their methods are intended to reduce or understand variance. Could the authors provide a citation for this claim? If no direct claims about variance reduction exist in the cited works, could the authors discuss in the work why they believe these methods should be evaluated in terms of variance reduction.

(c) While I agree with the authors that generally, subsampling might lead to critical issues in benchmarking – the results presented seem underwhelming given I fully buy the claim of using continuous metrics to reduce variance.

Specifically, the results in Table 7 indicate that Kendall’s tau still remains quite high and relatively stable. Do the authors believe this is damning evidence against efficient benchmarking methods? (I agree in principle there will be cases where this is damning, but the shown results do not seem to be those cases.) Similarly, Table3 and Table 4 show variance and monotonicity increases but the increase in variance/decrease in monotonicity remains quite small post-subsampling compared to the relative gains in benchmarking efficiency (except in the case of GSM-8k discrete metric).

**P3: MMLU Evaluation and Metric Selection**

I fully agree that continuous metrics may indeed have lower variance than discrete metrics (Table 2 demonstrates convincing gains in PSNR), the emphasis on MMLU in this context feels misplaced as one metric is near random.

Metrics that perform near random chance are unreliable indicators of progress, while those consistently above random are more dependable. However, for smaller models, this reliability may simply stem from selecting a metric that surpasses random performance earlier in training, rather than addressing the broader variance concerns noted above. I don't know why this specific example was picked.

P4. **Poor Writing**

The writing lacks clarity and precision, with ambiguous and poorly articulated claims scattered throughout the paper. This made the review difficult. In the summary, I’ve tried my best to interpret the paper’s main claims — please let me know if my summary underclaims compared to the intended or the claims differ from those presented in the work.

**Questions:**

See weaknesses above please.

Overall, I think we should definitely report variance in benchmark estimates to compare significance of improvement, however I believe retraining $k$ times to obtain seed variance might not be the critical factor in total variance, and hugely expensive. I do think the recommendation of using continuous metrics makes a lot of sense.

I think there are important shortcomings in my view, although I might likely be wrong. If weakness 1 is adequately alleviated, I would upgrade my score. Note: Weakness 3 and 4 are minor and have little effect on rating.

---

> ### Author Response · Authors · 2024-11-25
> **Response to Reviewer VsPx**
>
> We thank the reviewer for their comments and address the questions/feedback below:
>
> **(Part 1) Seed Variance and Its Impact on Benchmark Scores**
> > Does seed variance contribute overwhelmingly to the overall variance which makes it a critical factor to study, beyond [1,2,Arena]? I would like a better distinction in the work for why studying seed variance specifically is important, given the existing literature on other variance sources.
>
> We agree with the reviewer that the focus in this paper is on seed variance. But, this variance is quite an important factor for evaluations done during the early stages of pre-training. For example, experiments and ablations performed at different compute (FLOPs) budgets to build scaling laws, selecting the best data mixture and weights for pre-training, model architecture ablations. High variance during the early stages of pre-training and at smaller FLOPs budgets can lead to inaccurate selection of architecture/datamix leading to waste of significant amounts of computational resources for larger pre-training runs.
>
>
> **(Part 2) Efficient Benchmarking in High-Variance Contexts**
> > I doubt those papers claim their methods are intended to reduce or understand variance. Could the authors provide a citation for this claim? If no direct claims about variance reduction exist in the cited works, could the authors discuss in the work why they believe these methods should be evaluated in terms of variance reduction.
>
> The tinyBenchmarks paper (https://arxiv.org/pdf/2402.00838) motivate the use of IRT methods for pre-training ablations in Sections 1 and 5, which in hindsight may not be effective due to increased variance (Figure 5 shows this clearly). Through our analysis, we want to highlight that any kind of efficient benchmarking (tinyBenchmarks: https://arxiv.org/pdf/2402.00838, Mixeval: https://arxiv.org/abs/2406.06565, SMART: https://arxiv.org/abs/2410.20245) is limited in its use for pre-training ablations, building scaling laws, etc. especially during the early stages of pre-training because they exhibit high variance and hence are not reliable to distinguish between the various experimental settings which will transfer to larger models as well.
>
> > While I agree with the authors that generally, subsampling might lead to critical issues in benchmarking – the results presented seem underwhelming given I fully buy the claim of using continuous metrics to reduce variance. Specifically, the results in Table 7 indicate that Kendall’s tau still remains quite high and relatively stable. Do the authors believe this is damning evidence against efficient benchmarking methods?
>
> In Table 7, the last two columns are the most relevant where the percentage change in flips can be up to 30.77%. This shows that the efficient benchmarking methods built using weaker models are not transferable to more capable models, and can lead to inaccurate estimation of the actual final scores on the full evaluation test set.

---

### Official Review · Reviewer_7Zuc · 2024-11-03

**Soundness:** 3
**Presentation:** 3
**Contribution:** 2
**Rating:** 5
**Confidence:** 3

**Summary:**

This paper investigates the variance present in current evaluation benchmarks by repeating the training of models with different random seeds. It demonstrates the variance caused by random factors in benchmarks, providing valuable references for assessing model evaluation results, particularly for the MMLU evaluation of smaller-scale models. The paper attempts to reduce evaluation variance using methods inspired by human testing literature, such as item analysis and item response theory, but finds that these methods have limited effectiveness.

**Strengths:**

- By training different models from scratch, the paper provides the most direct results for evaluating the variance brought by randomness. Based on these results, it offers valuable suggestions for assessing smaller-scale models.
- The paper introduces methods from human testing literature and explains the reasons for their limited success, providing insights for future work.

**Weaknesses:**

- The paper focuses only on the variance caused by model random seeds and does not compare or combine this with existing work on the analysis of evaluation result variance. For example:
  1. It does not compare the variance caused by random seeds with the impact of other random factors during evaluation (such as option order, prompts, etc.).
  2. It does not explore whether the variance is further amplified when models with different seeds encounter situations like randomized options.
- The paper primarily showcases the overall variance of 210 7B model checkpoints. Given that models of different sizes exhibit significant differences in performance when trained with varying numbers of tokens, the overall variance statistics may have limited reference value for models trained with fewer or more data.

**Questions:**

- Considering that the author trained different models from scratch, such as Hellaswag, which consistently outperformed random choice across a wide range of checkpoints, it would be beneficial if the author could use intermediate checkpoints to demonstrate the benchmark variance at different stages of training and performance. Furthermore, showing how benchmark variance changes with training progress could be helpful for models of various sizes and training data volumes.
- When calculating Seed Variance, would it be more reasonable to exclude checkpoints that are clearly still within the random result range from the statistical analysis?
- When models of different sizes achieve the same performance after being trained with different numbers of tokens (e.g., a 7B model trained with 120B tokens and a 1.5B model trained with 400B tokens), do they exhibit significant differences in benchmark variance?

---

> ### Author Response · Authors · 2024-11-25
> **Response to Reviewer 7Zuc**
>
> We thank the reviewer for their comments and address the weaknesses/questions below:
>
> > It does not compare the variance caused by random seeds with the impact of other random factors during evaluation (such as option order, prompts, etc.). It does not explore whether the variance is further amplified when models with different seeds encounter situations like randomized options.
>
> It’s true that we don’t consider other factors like decoding strategies, temperature, prompt variations, etc. But we would like to add that this has little impact on log-likelihood based evaluations (in which no model generation is involved). The possible choices/text completions are appended to the prefix prompt and the option with the lowest NLL is chosen as the model response. In our paper, this corresponds to AGIEval, ARC-C, COPA, Hellaswag, MMLU, PIQA, and SIQA. Moreover, Dubey et al. 2024, https://arxiv.org/abs/2407.21783 (Figure 14) show that the prompt variations and option orders are fairly robust and exhibit negligible variance in this setup of NLL-based evaluation.
>
> > The paper primarily showcases the overall variance of 210 7B model checkpoints. Given that models of different sizes exhibit significant differences in performance when trained with varying numbers of tokens, the overall variance statistics may have limited reference value for models trained with fewer or more data.
>
> We train 7B models for 210B tokens accounting for a FLOPs budget of $10^{22}$, which is a fairly standard budget for building scaling laws. For example, Dubey et al., 2024 (Llama 3: https://arxiv.org/pdf/2407.21783) and Hoffman et al., 2022 (Chinchilla: https://arxiv.org/pdf/2203.15556). We believe that the reference variance values are representative of this budget: Table 2 column 3 - Disc Std for $10^{22}$ budget, and Table 1 column 5 - $\sigma(\mathcal{S}, \mathbb{M})$ for all budgets $\leq 10^{22}$.
>
> > Considering that the author trained different models from scratch, such as Hellaswag, which consistently outperformed random choice across a wide range of checkpoints, it would be beneficial if the author could use intermediate checkpoints to demonstrate the benchmark variance at different stages of training and performance. Furthermore, showing how benchmark variance changes with training progress could be helpful for models of various sizes and training data volumes.
>
> We do report the variance change over the course of training for all benchmarks in the paper in Figures 1, 2, and 6 for both discrete and continuous metrics. The y axis reports the performance metrics, and the box plots at each step show the variance corresponding to that step.
>
> > When calculating Seed Variance, would it be more reasonable to exclude checkpoints that are clearly still within the random result range from the statistical analysis?
>
> We don’t believe that would help as the LLM community is still using numbers that are near chance to compare performance across different models, for example, MMLU comparisons are done in https://arxiv.org/pdf/2310.04564, https://arxiv.org/pdf/2401.02385, https://arxiv.org/pdf/2312.06550, etc. but the performance is near chance.

---

> > ### Comment · Reviewer_7Zuc · 2024-11-28
> >
> > Thanks for the author's response. My questions regarding variance and the variance of intermediate checkpoints have been answered.
> >
> > I would like to further explain a key concern I mentioned earlier:
> > - *"The paper primarily showcases the overall variance of 210 7B model checkpoints. Given that models of different sizes exhibit significant differences in performance when trained with varying numbers of tokens, the overall variance statistics may have limited reference value for models trained with fewer or more data."*
> > - *"When models of different sizes achieve the same performance after being trained with different numbers of tokens (e.g., a 7B model trained with 120B tokens and a 1.5B model trained with 400B tokens), do they exhibit significant differences in benchmark variance?"*
> >
> > These questions arise from my concern that the variance reported for the 7B scale model may not be representative. For instance, in my own experiments, I found that two settings with similar FLOPs, such as a 7B model trained with 120B tokens and a 1.5B model trained with 400B tokens, showed that the 7B model had a better average benchmark performance. This indicates that different models can perform differently under the same FLOPs budget. The authors did not emphasize that the 7B model trained with 210B tokens is the optimal setting for the FLOPs budget discussed in the paper. While I understand the authors' point that the current experimental FLOPs is a fairly standard budget for building scaling laws, I still have concerns about the statement *"We believe that the reference variance values are representative of this budget."* Even in the process of modeling scaling laws, models trained with the same FLOPs can exhibit significant performance differences. How can this model size and training tokens represent all scales of experiments within the FLOPs budget?

---

### Official Review · Reviewer_V3wm · 2024-11-03

**Soundness:** 2
**Presentation:** 3
**Contribution:** 3
**Rating:** 6
**Confidence:** 3

**Summary:**

This paper aims to quantify the amount of variance prevalent in popular LLM evaluation benchmarks, mainly by varying the random seeds for model training. The paper makes an important point about how we currently consider benchmark scores to only be point estimates rather than considering several other different factors of stochasticity while doing model comparisons. The paper conducts several experiments to quantify this seed variance across different benchmarks, and provides practical guidance on what metrics and evaluations to use during pretraining to provide most signal.

**Strengths:**

- The paper presents a timely and important topic of variance in evaluation benchmarks, that should be widely considered while reporting benchmark performance.
- The paper is cogently written and presents convincing demonstrations of the importance of considering variance in evaluations while doing model comparisons.
- The paper showcases results and cautions against using sample efficient benchmarking methods while doing model pretraining, since they are likely to provide a higher-variance signal during training.

**Weaknesses:**

- The provided variance numbers in Tab. 1, while being important as a reference, cannot be used directly for making comparisons across different model scales or training durations since it is not clear how those numbers would change with those factors, and whether we’d expect larger or smaller deviations in performance.

- Some important empirical details are missing, for example, could you provide more details on how you compute the SNRs for both discrete and continuous? This is important since you are making the claim that Cont SNRs being larger than Disc SNRs suggests that we should shift towards continuous metrics for making model comparison decisions, and this claim can only be validated if the precise method of computing these SNRs is justified.

- Another important question that is critical for the takeaways of the paper: Is the unnormalised monotonocity the best metric here to capture the “stableness” of a benchmark? Shouldn’t that monotonicity be weighted by the monotonicity you would expect by chance? So something like a cohen’s kappa coefficient here seems more appropriate rather than just the direct unnormalised monotonicity. For eg, see the analysis in Geirhos et al, Beyond accuracy: quantifying trial-by-trial behaviour of CNNs and humans by measuring error consistency. I appreciate that this might be hard to formalise since its unclear what the “monotonoicty expected by chance is” but I believe this is worth at least a discussion point, and would like to hear the author’s thoughts on this. Also worth mentioning here that monotonicity was also explored for doing benchmark selection as good validation datasets for the fine-web training recipe. The FineWeb Datasets: Decanting the Web for the Finest Text Data at Scale.

- Suggestion on formatting the paper: the results sections in 3 and 4 have some very important insights / takeaways for practitioners to adopt for pretraining and actual evaluation. I think it would majorly improve the paper if these could be highlighted at the end of each subsection in bold, or even better, a small box signifying the key takeaway.

- There are other kinds of variance inducing factors that haven't been investigated. For example, how does task type affect this variance?
How does number of shots and choice of shots affect variance? How does model size impact variance? Does it increase as we scale up the model? There should be a discussion added about the other variance inducing factors which haven't been considered in the current work, and it must be made clear in the paper that "seed variance" is the only type of variance being investigated in this work.

**Questions:**

I have quite a few questions that I think would improve the quality of the paper, and for my own clarifications:

- Why are the Disc Std in tab 2 and std in tab 1 different? Aren’t they computing the same metric over the same set of models? The means are exactly the same for the discrete metrics so I presume the stds should be too?

- In tab. 2 what decoding strategy do you use for the generative tasks? The difference between GSM8k (0.99) and HumanEval (0.21) seems quite high for a difference in log-likelihoods. Are these token-length normalised likelihoods or unnormalised likelihoods?

- Comment: Typo in fig 3 caption, should be item difficulty (y-axis) and item discriminability (x-axis).

- For the analysis in fig 3, how does the correlation between item discrimination b/w train and test look like for another randomly selected set of train and test models? The key question is what the variance in the correlation obtained for a random split of random train-test models would be? This would make the conclusion (that the low correlation between train and test item discriminability is due to the differences in model capability) more strong and robust.

- The “best" and “worst” models used for creating the train and test splits do not share the same training data mixtures right? So I would expect that the takeaways from fig 3, especially the ones involving the difficulty split are also confounded by different levels of data contamination with respect to the test sets?

- For the points in section 5, the main claims revolve around using the IRT benchmarks themselves. How much of that can be explained by the increased variance from just having a smaller number of test samples? i.e. how would the results look like if instead of the IRT test set, a random set of evaluation points of the same size as the IRT set were used for the analyses?

---

> ### Author Response · Authors · 2024-11-25
> **Response to Reviewer V3wm**
>
> We thank the reviewer for their thoughtful comments and address the weaknesses and questions below:
>
> > Some important empirical details are missing, for example, could you provide more details on how you compute the SNRs for both discrete and continuous?
>
> We compute SNR using the following formula, $\frac{\mu(\mathcal{S}, \mathbb{M}^{210B})}{\sigma(\mathcal{S}, \mathbb{M}^{210B})}$. We’ve updated it in the revised version.
>
> > Another important question that is critical for the takeaways of the paper: Is the unnormalised monotonocity the best metric here to capture the “stableness” of a benchmark? Shouldn’t that monotonicity be weighted by the monotonicity you would expect by chance? So something like a cohen’s kappa coefficient here seems more appropriate rather than just the direct unnormalised monotonicity. For eg, see the analysis in Geirhos et al, Beyond accuracy: quantifying trial-by-trial behaviour of CNNs and humans by measuring error consistency.
>
> It’s very hard to quantify the monotonicity expected due to chance as it’s not evidently clear what mean value to take over the course of time. If we just treat each step independently, then the chance performance is the same across all steps, resulting in a monotonicity of 0. But obviously, it’s not independent and develops for each benchmark differently depending on the “hidden” scaling laws for that benchmark. There’s a wealth of literature pointing out the defects with Cohen's kappa coefficient (for e.g. Appendix B of https://arxiv.org/abs/2406.12624 for a short summary).
>
> As for Geirhos et al, Beyond accuracy: quantifying trial-by-trial behaviour of CNNs and humans by measuring error consistency, can the reviewer expand on how the error consistency analysis can be used for monotonicity?
>
> > Suggestion on formatting the paper: the results sections in 3 and 4 have some very important insights / takeaways for practitioners to adopt for pretraining and actual evaluation. I think it would majorly improve the paper if these could be highlighted at the end of each subsection in bold, or even better, a small box signifying the key takeaway.
>
> We agree with the reviewer and will update the manuscript accordingly.
>
>
> > There are other kinds of variance inducing factors that haven't been investigated. For example, how does task type affect this variance? How does number of shots and choice of shots affect variance? How does model size impact variance?
>
> We agree that there are other sources of variance as well, however, for pre-training ablations involving scaling laws and datamix selection, seed variance is an important factor of variance. We’ll add a discussion on other sources of variance while highlighting that this paper considers "seed" variance.
>
> > Why are the Disc Std in tab 2 and std in tab 1 different? Aren’t they computing the same metric over the same set of models? The means are exactly the same for the discrete metrics so I presume the stds should be too?
>
> For the means computed in both Table 1 and 2, we use the final checkpoints across the different seed runs as defined in Lines 123-124. However, for the standard deviation in Table 1, we use the seed variance $\sigma(\mathcal{S}, \mathbb{M})$ definition provided in Lines 126-133, taking into account intermediate checkpoints as well. For the standard deviation in Table 2, we use only the final checkpoints across seeds for an accurate assessment of the signal to noise ratio. That’s why we use Disc Std to distinguish from the seed variance $\sigma(\mathcal{S}, \mathbb{M})$.
>
> > In tab. 2 what decoding strategy do you use for the generative tasks? The difference between GSM8k (0.99) and HumanEval (0.21) seems quite high for a difference in log-likelihoods. Are these token-length normalised likelihoods or unnormalised likelihoods?
>
> We use greedy decoding (temperature $= 0$) for sampling and use the character-length normalized NLL for both GSM8k and HumanEval. The token-length normalized NLLs also exhibit similar differences as shown in Figure 8 (Section C.2) in the updated version of the paper.
>
> > Comment: Typo in fig 3 caption, should be item difficulty (y-axis) and item discriminability (x-axis)
>
> Thanks for pointing it out, we have fixed it in the revised version of the paper.

---

> ### Author Response · Authors · 2024-11-27
> **Response to Reviewer V3wm (continued)**
>
> > For the analysis in fig 3, how does the correlation between item discrimination b/w train and test look like for another randomly selected set of train and test models? The key question is what the variance in the correlation obtained for a random split of random train-test models would be? This would make the conclusion (that the low correlation between train and test item discriminability is due to the differences in model capability) more strong and robust.
>
> We have updated the paper with additional results. In Figure 10, you can see that for five different sets of train/test splits (with 14 models each in the test sets), the results are robust and the regression line fits are similar with high correlation on item discrimination between the train and test models in the random splits. This enforces the claim of low correlation in the train and test item discriminability is actually due to differences in model capability.
>
> > The “best" and “worst” models used for creating the train and test splits do not share the same training data mixtures right? So I would expect that the takeaways from fig 3, especially the ones involving the difficulty split are also confounded by different levels of data contamination with respect to the test sets?
>
> Yes, the reviewer is correct. Since many of these models are just open-weights, we don't know the level of contamination of test sets in their pre-training datamixes. Assuming that the folks releasing the models are fair players and that the performance difference for a given pair of models accurately reflects model capabilities, we believe our findings still hold.
>
> > For the points in section 5, the main claims revolve around using the IRT benchmarks themselves. How much of that can be explained by the increased variance from just having a smaller number of test samples? i.e. how would the results look like if instead of the IRT test set, a random set of evaluation points of the same size as the IRT set were used for the analyses?
>
> Since the optimization objective for IRT benchmarks is to preserve the mean performance and taking any random sample won't preserve it, we believe it's not an accurate comparison between the two cases, and the random sample would not give any kind of signal as provided in Figure 5. Our goal for section 5 is to highlight that any efficient benchmarking in general (and not just IRT) can be very noisy and lead to increased variance during the scaling laws/pre-training ablations, and not effective at all compared to the full evaluation.

---

### Official Review · Reviewer_i8dM · 2024-11-04

**Soundness:** 2
**Presentation:** 2
**Contribution:** 2
**Rating:** 5
**Confidence:** 3

**Summary:**

This paper discusses the problem in current evaluation and reporting practices where performance may vary across different development choices. This variance is scoped for training initialization seed and monotonicity. The paper also discusses ways to reduce this variance, in particular for choice based benchmarks.

**Strengths:**

- The paper explores how to make evaluations more precise by reporting variance.
- Provides estimates for the expected variance across several benchmarks and models.
- An important finding is made on the unreliability of IRT-based methods for evaluation comparisons across models. This is very relevant for evaluation reporting.

**Weaknesses:**

- The framing of ‘variance’ in the paper seems too broad. There are other possible kinds of variance worth exploring or mentioning.
- The title of the paper and general framing suggests a general focus on variance in evaluations, but the paper currently fails to contextualize two very distinct types of variance: training and inference. For example, the (training) seed variance discussed falls within training. Other possible sources of variance for each should be mentioned where possible. A basic example of the inference type can be found in prompt sampling.
- While studying the training seed variance is useful, this is really only feasible for smaller models, as it would be too expensive for larger models. This may reduce the utility of the results in large model comparisons.

The paper could mention previous work such as Picard 2021 (https://arxiv.org/abs/2109.08203) on the impact of training initialization seed variance, or fine tuning seed variance [Dodge 2020 (https://arxiv.org/abs/2002.06305)]. And also extended discussions on how obscuring or not disclosing these variances can be harmful to the evaluation process (e.g. [Leech 2024 (https://arxiv.org/abs/2407.12220)]).

**Questions:**

- Are there any other kinds of variance that could have been used for this study instead of initialization seed?
- Only training-based sources of variance are discussed, what about inference-based?
- Section 3.3 outlines a very interesting case of differences in evaluation results after a reformulation of the setup, could this be shown for other benchmarks? perhaps a similar pair?

---

> ### Author Response · Authors · 2024-11-25
> **Response to Reviewer i8dM**
>
> We thank the reviewer for their comments on our paper and respond to their questions below:
>
> > The framing of ‘variance’ in the paper seems too broad. There are other possible kinds of variance worth exploring or mentioning.
>
> We agree with the reviewer that there are other sources of variance, however the focus in this paper is on variance in evaluations during pre-training, where seed variance is an important factor in building scaling laws, and variance as a consequence of the fact that eval sets are finite samples of all questions that could be asked impact how we should interpret their results and when differences are statistically significant.
>
> > While studying the training seed variance is useful, this is really only feasible for smaller models, as it would be too expensive for larger models. This may reduce the utility of the results in large model comparisons.
>
> We kindly disagree with this assessment because before any large scale pre-training runs for a huge model, there’s a lot of investment on small scale experiments to build the scaling laws, decide the model architecture or the pre-training datamix, etc. The small scale experiments are usually performed on a lower FLOPs budget, where the variance can play a significant role. High variance can cause inaccurate selection of model architecture/datamix that hurt the large-scale pre-training runs.
>
> > The paper could mention previous work such as Picard 2021 (https://arxiv.org/abs/2109.08203) on the impact of training initialization seed variance, or fine tuning seed variance [Dodge 2020 (https://arxiv.org/abs/2002.06305)]. And also extended discussions on how obscuring or not disclosing these variances can be harmful to the evaluation process (e.g. [Leech 2024 (https://arxiv.org/abs/2407.12220)]).
>
> We thank the reviewer for the extra references, and will make sure to include this in the updated version of the paper.
>
> > Section 3.3 outlines a very interesting case of differences in evaluation results after a reformulation of the setup, could this be shown for other benchmarks? perhaps a similar pair?
>
> Yes, this is applicable to other benchmarks as well, especially those that have a MCQ-based evaluation setup. The model learns the ability to answer MCQ-based questions later in training, and using a cloze format is helpful in tracking performance in the early stages of pre-training.

---

### Official Review · Reviewer_tZvi · 2024-11-05

**Soundness:** 3
**Presentation:** 3
**Contribution:** 1
**Rating:** 3
**Confidence:** 3

**Summary:**

This work aims to quantify evaluation benchmark variance across a range of settings (from pretraining intermediate checkpoints, to the largest frontier LLMs) using a diverse set of metrics (seed variance, confidence intervals, and monotonicity). Beyond quantifying variance, the paper also explores item response theory.

**Strengths:**

The paper is clearly written and easy to understand.

**Weaknesses:**

The key problem for me is that I do not get the value proposition of this work. It's difficult to see how this work could develop relevance/impact for the evaluation of foundation models. Since that's why I am hesitant to support the paper I focus my review and questions below fully on this point.

**Questions:**

1. What is the most striking example with which you can demonstrate the potential impact of this work?
2. There are many challenges to evaluating foundation models and it is clearly also a matter of how much money (time, compute) to invest in which aspect to arrive at a conclusive result. So evaluation is inherently a trade-off and it is important to understand and acknowledge this trade-off. In my opinion there seem to be much more critical aspects that need to be addressed than the variance studied in this paper. For example, the paper "EVALUATING LLMS’ MATHEMATICAL AND CODING COMPETENCY THROUGH ONTOLOGY-GUIDED INTERVENTIONS" by Pengfei Hong et al seems to be a good route towards useful evaluations. I would rather invest time and compute into that direction and not bother about the methods proposed in this submission. Are you advocating for the opposite strategy?

---

> ### Public Comment · ~Clarence_Lee3 · 2024-11-25
> **Supporting the Practical Significance of this Work in LLM Pretraining**
>
> I am not an author of this paper, nor am I affiliated with the authors. I am an independent practitioner with significant experience in LLM pretraining development. I respectfully disagree with the reviewer's assessment regarding the impact of this work and would like to provide additional perspective.
>
> Discerning reliable evaluation signals in model experiments remains a critical challenge in LLM pretraining. One particular impact is in the performance of data ablations (as described in Section 3 of the paper). For instance, it is common to employ grid search experiments to compare various data mixtures during pretraining. However, a persistent challenge lies in determining whether observed evaluation score differences represent meaningful improvements or are merely artifacts of noise (Even when one data mixture yields better scores, it is not always clear if this reflects a better training mixture for training). This challenge is further compounded by the practical constraint that such experiments typically operate at smaller pretraining scales (e.g., 10–50 billion tokens, as opposed to trillions), which inherently reduces the statistical signal and makes conclusions harder to draw.
>
> The paper’s approach of pretraining models using multiple random seeds to significant token counts is a commendable effort to address this problem. While logistically demanding, this methodology is essential for producing reliable, reproducible insights. Few papers in the LLM field invest in such rigorous experimentation, which makes this work both novel and highly valuable.
>
> In response to Reviewer tZvi's second point: while ontology-guided evaluations may offer a structured way to analyze models, they do not address the critical issue of variance in evaluation outcomes (for example, ontology guided evaluations does not guarantee meaningful signals across different data mixture or hyperparameter experiments). This variance often obscures meaningful signals and poses significant challenges for practical application. The real problem lies not only in designing better evaluations but also in understanding how evaluation performance correlates with different training dynamics. This paper directly tackles this overlooked yet vital aspect.
>
> The suggestion to deprioritize research in this area reflects a potential misunderstanding of its relevance to the LLM pretraining community. As someone actively engaged in this domain, I can affirm that the contributions of this paper are both impactful and timely. Its focus on robust evaluation methodologies fills a critical gap in the literature, addressing challenges that practitioners encounter frequently. Moreover, by pretraining models from scratch and explicitly reporting variances, the authors provide actionable insights that are useful for research in this space.
>
> I strongly encourage the reviewers to reconsider their assessment of this paper’s impact, given the relevancy and impact towards more robust LLM pretraining

---

> ### Author Response · Authors · 2024-11-25
> **Response to reviewer tZvi**
>
> We thank the reviewer for their time, but we are disheartened by the reviewer's comments on the value of our work and their assessment on the potential impacts of our paper.
>
> > What is the most striking example with which you can demonstrate the potential impact of this work?
>
> One of the best open-weights models (Llama 3: https://arxiv.org/pdf/2407.21783) uses confidence intervals in their performance score reporting, while addressing the variance problem. More recently, researchers from Anthropic (https://www.anthropic.com/research/statistical-approach-to-model-evals) suggest approaches like Central Limit Theorem in computing the error estimates for evaluation.
>
> Moreover, even though these approaches are fairly standard, the LLM community seldom uses this in reporting performance scores across a range of tasks. We suggest a positive step in that direction by analyzing the variance arising from the "seed" used in pre-training models. We also do a comprehensive analysis on the pitfalls of efficient benchmarking which compounds the variance, resulting in inaccurate performance estimates for models.
>
> > There are many challenges to evaluating foundation models and it is clearly also a matter of how much money (time, compute) to invest in which aspect to arrive at a conclusive result. So evaluation is inherently a trade-off and it is important to understand and acknowledge this trade-off. In my opinion there seem to be much more critical aspects that need to be addressed than the variance studied in this paper. For example, the paper "EVALUATING LLMS’ MATHEMATICAL AND CODING COMPETENCY THROUGH ONTOLOGY-GUIDED INTERVENTIONS" by Pengfei Hong et al seems to be a good route towards useful evaluations. I would rather invest time and compute into that direction and not bother about the methods proposed in this submission. Are you advocating for the opposite strategy?
>
> We disagree with the reviewer's assessment and as the kind stranger pointed out in the other comment, variance in evaluations during pre-training hinders the ablations and experiments performed for building scaling laws, selecting datamixes for pre-training, etc. If there's significant variance, it's hard to discriminate between the various experimental settings and one might end up selecting a datamix or architecture which hurts performance over the course of a full pre-training run, leading to significant waste in computational resources.

---

> > ### Comment · Reviewer_V3wm · 2024-11-27
> > **Comments on reviewer concerns and author rebuttal**
> >
> > On the second point raised by reviewer tZvi and the authors' response: I disagree with the reviewer's assertion that the methods proposed in this paper along with the general idea of quantifying variance in the manner studied in the paper, are not useful. I completely align with the authors' rebuttal, especially on the following point:
> >
> > > If there's significant variance, it's hard to discriminate between the various experimental settings and one might end up selecting a datamix or architecture which hurts performance over the course of a full pre-training run, leading to significant waste in computational resources.
> >
> > As a practitioner myself, I have faced this same exact issue multiple times and completely agree with the authors that a lack of systematic knowledge about the variance of particular evaluations can cause significant wastage in terms of compute and resources due to getting flawed signals---by making design choices based on high-variance evaluations and pursuing/discarding different methods based on these flawed signals, several promising methods could be wrongly discarded simply due to looking at the wrong evaluation metrics. In-fact, such approaches of only including low-variance evaluations have also been followed by recent foundation model training runs by [Huggingface](https://huggingfacefw-blogpost-fineweb-v1.static.hf.space/dist/index.html#ablations_and_evaluation_setup) and [DatologyAI](https://www.datologyai.com/post/productionized-multimodal-data-curation-at-the-billion-sample-scale). Hence, I believe quantifying the exact variance of different evaluations is highly important.
> >
> > Given this evidence, I completely back the authors' view on this.

---

### Meta-Review · Area_Chair_DT9w · 2024-12-19

**Metareview:**

This work investigates the variance in LLM evaluation benchmarks, primarily looking at sources of variance that manifest during training rather than at test time. To this end, a set of 7B Llama-style models are trained and experiments are conducted to determine how the conclusions drawn from benchmarks can vary throughout the training process. Various factors are explored in the experiments, including whether discrete or continuous metrics are used, how choice tasks are framed, and whether techniques from human testing can reduce variance.

The reviewers mostly had a negative opinion of this paper, with the bulk of the criticism centred on the perceived significance of variance due to the choice of random seed/initialisation. Several reviewers (i.e., tZvi, i8dM, 7Zuc, and VsPx) suggested that it would be more worthwhile to investigate other sources of variance that manifest at inference time. I have a dissenting opinion here: the focus of this submission is on how variance in benchmarking *during pre-training* can impact subsequent modelling choices. This is in contrast to much of the other work in this area and the suggestions of the reviewers, who tend to focus on evaluating models that have already finished training. With that said, I still am not prepared to override the majority decision of the reviewers. As pointed out by reviewer w3M8 and V3wm, the submission in its current form does not allow for inferences to be made across model scales, which would be important for deriving scaling laws that are robust to evaluation variance during training.

**Additional Comments On Reviewer Discussion:**

There was quite a bit of discussion related to which sources of variance are worthy of study, with the authors and several reviewers not coming to a consensus.

I would also like to take this opportunity to agree with Clarence Lee and reviewer V3wm, that the main weakness identified by reviewer tZvi is not well-founded.

---

### Decision · Program_Chairs · 2025-01-22

Reject